# EAMET: ROBUST MASSIVE MODEL EDITING VIA EMBEDDING ALIGNMENT OPTIMIZATION

**Yanbo Dai, Zhenlan Ji, Zongjie Li\*, Shuai Wang\***
Department of Computer Science and Engineering
The Hong Kong University of Science and Technology
`{ydai851, zjiae, zligo, shuaiw}@cse.ust.hk`

## ABSTRACT

Model editing techniques are essential for efficiently updating knowledge in large language models (LLMs). However, the effectiveness of existing approaches degrades in massive editing scenarios, particularly when evaluated with practical metrics. Their robustness is also limited in context-rich settings or when editing multiple facts of the same subject simultaneously. We attribute these failures to the embedding misalignment among knowledge items, which undermines editing reliability at scale. To address this, we propose EAMET (Embedding Alignment Model Editing in Transformers), which addresses this issue by aligning the space of key and residual embeddings. Extensive experiments across six LLMs and three datasets demonstrate that EAMET consistently outperforms existing methods, achieving about 90% editing efficacy when editing 10k facts. Codes and datasets are publicly available at `https://github.com/ybdai7/EAMET-massive-editing`.

## 1 INTRODUCTION

Large language models (LLMs) are increasingly employed as search engines and chatbots, as they excel at retrieving knowledge to answer user queries (Brown et al., 2020; Touvron et al., 2023b; Yang et al., 2024a; Bi et al., 2024). However, they are prone to spreading misinformation about frequently updated topics due to outdated training data (Vykopal et al., 2023; Huang et al., 2025; Xu et al., 2024). To address this issue, retraining or fine-tuning models for partial knowledge updates is proposed (Achiam et al., 2023; Team et al., 2025), albeit with prohibitively expensive overhead. In contrast, recent advances in locate-then-edit model editing (ME) techniques (Meng et al., 2023; Fang et al., 2024) enable massive editing of thousands of factual associations concurrently at minimal data and computational cost, thereby rendering real-time knowledge updates feasible.

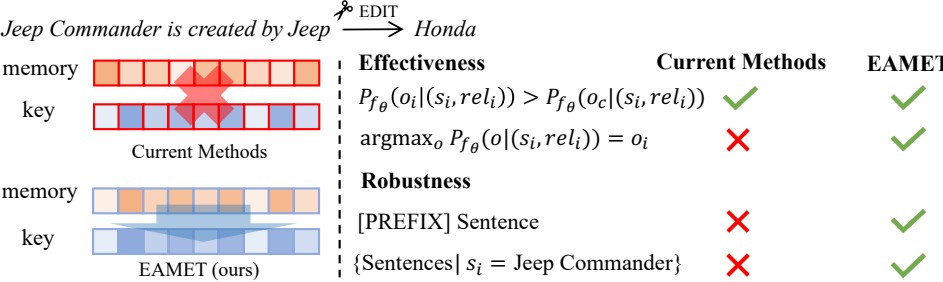

Figure 1: Illustration of current methods and our proposed EAMET in evaluating massive editing. Here, "[PREFIX] Sentence" and "{Sentence | $s_i$ = Jeep Commander}" denote the scenarios where the edited knowledge is preceded by prefixes and where multiple facts share the same subject, respectively.

Despite the success of existing massive ME techniques, we observe that their *effectiveness* is often overestimated due to overly loose evaluation metrics. In particular, most prior works assess editing

---

\*Corresponding authors.

quality by checking whether the model is *more likely* to generate the following tokens as the target object than the original one, whereas neglecting to evaluate whether the model's output is *consistent with the target object* (Meng et al., 2023; Fang et al., 2024). Therefore, we advocate a "practical metric", which measures the proportion of cases in which the edited model retrieves the target object and explicitly generates related output. This metric provides a more accurate reflection of real-world usage, as will be shown in our evaluation setting (see Section 6). Under such evaluation criteria, existing methods fail to maintain their performance.

Moreover, existing methods exhibit limited *robustness* in realistic settings. We highlight two representative scenarios: (i) their performance substantially degrades when edited knowledge is preceded by prefixes (Li et al., 2024), a common phenomenon in practical question-answering tasks (Pramanick et al., 2024; Romero et al., 2024); and (ii) they fail to preserve accuracy when editing multiple facts associated with the same subject, where performance drops markedly. Such lack of robustness in massive editing scenarios undermines their applicability to real-world use cases.

To analyze the limitations of existing methods, we first identify *"embedding misalignment"*, which reflects the structural inconsistency between key and residual embedding spaces, as a primary factor underlying the decline in both effectiveness and robustness during massive editing. Such misalignment leads to information loss for individual knowledge updates. In particular, when parameters are updated jointly from a batch of edited knowledge items, they fail to accurately reconstruct an individual factual association. This information loss becomes more severe as the number of edited items increases.

To achieve effective and robust massive editing under practical settings, we thus propose **EAMET** (**E**mbedding **A**lignment **M**odel **E**diting in **T**ransformers), which outperforms existing approaches under stricter evaluation criteria and exhibits strong robustness in two described scenarios. EAMET addresses embedding misalignment by progressively preserving optimized residual embeddings and aligning them with the key embedding space, ensuring consistency throughout the editing process.

In this paper, we conduct extensive experiments on six LLMs, showing that EAMET consistently surpasses existing methods under rigorous settings across the CounterFact, ZsRE, and Wiki-recent datasets. EAMET maintains about 90% editing efficacy across all evaluated models and outperforms baselines by an average of 14% and 8%, with gains of up to 37% and 15% on CounterFact and ZsRE when editing 10k facts. Moreover, EAMET sustains high accuracy even when edited items are preceded by prefixes of up to 200 tokens or involve multiple facts associated with the same subject. This demonstrates EAMET's robustness in realistic and context-rich settings, including chatbots and long-context QA tasks.

## 2 RELATED WORK

**Model Editing.** Existing ME techniques can be classified into auxiliary-based (Hartvigsen et al., 2023; Mitchell et al., 2022b; Zheng et al., 2023; Yu et al., 2024; Mitchell et al., 2022a) and location-based methods (Meng et al., 2022; 2023; Li et al., 2025). Auxiliary-based ME techniques preserve the original parameters, and introduce additional information to edit knowledge. SERAC (Mitchell et al., 2022b) requires extra memory to store new edits and learn to reason over them to manipulate the model's output. Location-based methods directly modify model parameters to edit knowledge without requiring any additional information. These methods assume that factual associations are stored in the feed-forward networks (FFNs) of the LLMs (Geva et al., 2021; 2022; Dai et al., 2022). Building on these, ROME (Meng et al., 2022) first gains insights on the specific location of the knowledge through causal analysis. It proceeds to directly modify critical MLP layers to update factual associations. MEMIT (Meng et al., 2023) builds upon ROME to enable massive editing of thousands of facts concurrently. AlphaEdit (Fang et al., 2024) focuses on sequential editing, aiming to preserve both previously edited knowledge and the general capabilities of the LLM during successive edits.

**Massive Editing.** In practical applications, ME techniques may aim to update a model with hundreds or even thousands of facts simultaneously in order to keep up with the constantly evolving knowledge (Ju et al., 2024; Gu et al., 2024). However, auxiliary-based methods are usually limited in scalability, typically supporting only a few edits at a time (Mitchell et al., 2022b). In contrast, location-based methods are more scalable for massive editing. MEMIT (Meng et al., 2023) scales to

edit 10,000 facts concurrently, and PMET (Li et al., 2025) further improves performance by incorporating attention layers when updating the parameters of the FFNs. Despite their effectiveness and scalability, these methods have been shown to be fragile when handling *prefixes* or multiple facts with the *same subject* during evaluation, which is a common scenario in real-world applications (Li et al., 2024; Yang et al., 2024b; Ma et al., 2024). Moreover, we observe that their performance in massive editing is overestimated due to the loose metric. In this work, we propose EAMET, which achieves superior performance in massive editing under practical evaluation metrics, while also exhibiting greater robustness against long prefixes and multiple facts with the same subject.

## 3  PRELIMINARY: EDITING MEMORY IN LLMS

Previous works have shown that a pre-trained LLM has memorized many factual associations (Petroni et al., 2019; Jiang et al., 2020; Roberts et al., 2020; Shin et al., 2020). These stored facts could be edited by modifying the MLP layers within FFN modules, based on the assumption that knowledge is stored in them in the form of key-value pairs (Geva et al., 2021; 2022).

In Figure 2, the MLP layer $W_{out}^l$ within FFN associates *keys* $k_t^l(x) = \sigma(W_{in}^l \gamma(h_t^{l-1}(x)))$ with *memories* $m_t^l(x)$ for the fact $x$. Given the critical mediating role of MLP layers in storing facts, Meng et al. (Meng et al., 2022) shows that it is sufficient to update $W_{out}^l$ to edit stored facts. We then optimize $W_{out}^l$ (abbreviated as $W_1$) as follows:

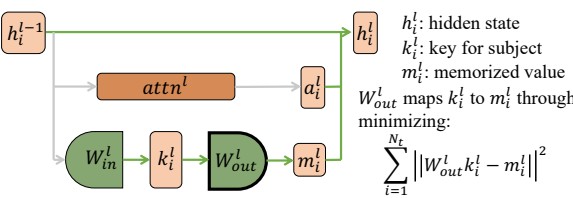

Figure 2: Illustration of the model editing problem.

$$W_1 \triangleq \arg\min_{\hat{W}} \left( \sum_{i=1}^{N_t} ||\hat{W}k_i^t - m_i^t||^2 + \sum_{j=1}^{N_p} ||\hat{W}k_j^p - m_j^p||^2 \right) \tag{1}$$

Here, $k_i^t$ and $k_j^p$ denote the encoded subject representations for individual target and preserved fact $i$ and $j$, respectively, while $m_i^t$ and $m_j^p$ represent their corresponding memory vectors. We stack the keys and memories of totally $N_t$ target knowledge into matrices as $K_t = [k_1^t \mid k_2^t \mid \ldots \mid k_{N_t}^t]$ and $M_t = [m_1^t \mid m_2^t \mid \ldots \mid m_{N_p}^t]$. Similarly, we construct $K_p$ and $M_p$ for $N_p$ preserved facts. The objective in Equation (1) can then be optimized by solving the normal equations (Meng et al., 2023):

$$(W_0 + \Delta)[K_p \quad K_t] = [M_p \quad M_t] \tag{2}$$

$$W_0 K_p = M_p \tag{3}$$

where we expand $W_1$ into $W_0 + \Delta$. $W_0$ denotes the original (unedited) parameters that associate preserved keys with their memory representations. The final update to $W_{out}^l$ can be computed by multiplying both sides of Equation (2) by $[K_p \quad K_t]^T$, and subtracting Equation (3) from Equation (2)(Meng et al., 2023):

$$\Delta(C_p + K_t K_t^T) = RK_t^T \tag{4}$$

where $R = M_t - W_0 K_t$ denotes new relations' residual with respect to the original weights, which can also be written as $[r_1^t \mid r_2^t \mid \ldots \mid r_{N_t}^t]$. Since the pretraining data of the original model is not accessible, we approximate $C_p$ using a set of randomly sampled inputs from public datasets:

$$C_p = \lambda E_{k^p}[k_i^p (k_i^p)^T] \tag{5}$$

The scalar $\lambda$ balances the influence between newly edited facts and preserved knowledge.

## 4  MOTIVATION

In this section, we investigate the root causes of the challenges associated with effective and robust massive editing, as illustrated in Figure 1. In particular, we analyze the decline in editing performance as the number of edited facts increases. Our theoretical and empirical results indicate that these issues arise from misalignment between key and residual embeddings. We further examine robustness in two representative scenarios: (i) *edits preceded by long prefixes*, and (ii) *edits applied to multiple facts sharing the same subject*.

## 4.1 EMBEDDING MISALIGNMENT IN EFFECTIVE MASSIVE EDITING

**Theoretical Analysis.** We observe that by expanding $K_t$ and $R$ in Equation (4), the update equation can be reformulated as:

$$\Delta\left(C_p + \sum_{i=1}^{N_t} k_i k_i^T\right) = \sum_{i=1}^{N_t} r_i k_i^T \tag{6}$$

where the update $\Delta$ is determined by the aggregated residual and key embeddings across all edited facts. As the number of edits increases, solving Equation (6) is more likely to cause reconstruction loss for individual knowledge items due to the *embedding misalignment* between the residual and key embeddings. This eventually leads to degraded editing performance.

To formalize the concept of embedding misalignment, we define two key requirements for the desired update $\Delta$: (1) The update should preserve the existing knowledge, expressed as $\Delta C_p = 0$. (2) The update should ensure lossless reconstruction for each individual fact, formulated as $\Delta k_i = r_i$, where $\Delta$ is computed while considering all target facts. Incorporating (1), an ideal $\Delta$ that meets (2) implies:

$$\Delta\left(C_p + \sum_{i=1}^{N_t} k_i k_i^T\right) = \sum_{i=1}^{N_t} r_i k_i^T \quad \rightarrow \quad \Delta k_i = r_i \quad \text{for } i = 1, 2, \ldots, N_t \tag{7}$$

However, the validity of Equation (7) is intuitively affected by the degree of misalignment between the residual and key embedding of different facts. We then define embedding misalignment:

**Definition 1 (Embedding Misalignment).** *Given $N$ knowledge items, let each item $i$ be associated with a residual embedding $r_i$ and a key embedding $k_i$. We define the embedding misalignment of item $i$ as the structural similarity between the pairwise relations of its residual embedding and those of its key embedding. Formally, consider the distributions*

$$P_r^{(i)} = \{ cos(r_i, r_j) \mid j \neq i \}, \qquad P_k^{(i)} = \{ cos(k_i, k_j) \mid j \neq i \}, \tag{8}$$

*where $cos(\cdot, \cdot)$ is the cosine similarity. The $i$th misalignment score is quantified by the KL divergence:*

$$\mathcal{A}(i) = \text{KL}\left(P_r^{(i)} \parallel P_k^{(i)}\right). \tag{9}$$

We now formalize the connection between embedding misalignment and the editing performance of a specific knowledge item $i$ under massive editing. Specifically, we quantify the degree to which Equation (7) is established by analyzing the reconstruction loss $e_i = \Delta k_i - r_i$ for each knowledge item. This relationship is formalized in the following theorem:

**Theorem 1.** Let $\Delta$ be the closed-form solution satisfying $\Delta \sum_i k_i k_i^\top = \sum_i r_i k_i^\top$, and define the reconstruction residual of item $i$ as $e_i = \Delta k_i - r_i$. Then we can expand

$$e_i = \sum_{j=1}^{N} \beta_{ij} r_j - r_i, \qquad \beta_{ij} := k_j^\top \Big(\sum_{\ell=1}^{N} k_\ell k_\ell^\top\Big)^{-1} k_i \tag{10}$$

and its norm is bounded by the misalignment between the neighborhood structures of $r_i$ and $k_i$:

$$\|e_i\| \leq C_i \sqrt{\tfrac{1}{2}\mathcal{A}(i)} + |\beta_{ii}| \|r_i\| + \|\varepsilon_i\|, \tag{11}$$

This result demonstrates how embedding misalignment impacts the editing performance of individual knowledge items under massive editing. Specifically, stronger misalignment among knowledge items leads to increased individual reconstruction loss, ultimately reducing the overall effectiveness of massive editing. The complete proof is provided in Appendix B.

**Empirical Study.** Motivated by the above analysis, we hypothesize that the failure of massive editing stems from misalignment between the embeddings of different knowledge items. To test this hypothesis, we edit 200, 500, and 1,000 facts from the CounterFact dataset (Meng et al., 2022) using MEMIT (Meng et al., 2023) on LLaMA2-7B (Touvron et al., 2023a) and Deepseek-7B (Bi et al., 2024). We then evaluate the editing accuracy of these items when no prefix is added to the edited query. Embedding misalignment is quantified using the misalignment score defined in Equation (9).

As shown in Table 1, the overall editing accuracy of both models decreases as more facts are edited, accompanied by a clear increase in embedding misalignment. For example, on LLaMA2-7B, the accuracy drops from 98.5% to 86.8% as the number of edited facts grows from 200 to

Table 1: Editing performance with varying numbers of edited facts on LLaMA2-7B and Deepseek-7B.

| Model | LLaMA2-7B | | | Deepseek-7B | | |
|---|---|---|---|---|---|---|
| No. of Edited Facts | 200 | 500 | 1000 | 200 | 500 | 1000 |
| Editing Efficiency(%) | 98.5 | 90.0 | 86.8 | 99.5 | 98.6 | 97.8 |
| $\sum_i \mathcal{A}(i)$ | 79 | 243 | 554 | 68 | 223 | 562 |

1,000, while the misalignment score rises from 79 to 554. These results provide further evidence for our theorem that embedding misalignment leads to degraded editing performance.

## 4.2 Impact of Embedding Misalignment on Editing Robustness

We investigate how embedding misalignment affects robustness in massive editing along two dimensions: (i) long-prefix perturbations and (ii) simultaneous edits of samples sharing the same subject. Based on our theoretical and empirical analysis, we derive two corollaries to characterize these effects and validate them with controlled experiments.

**Corollary 1.** *Long prefixes exacerbate embedding misalignment issues under massive editing, leading to degraded editing performance when edited facts are evaluated with descriptive prefixes.*

Table 2: Impact of varying prefix lengths on editing performance.

| Model | LLaMA2-7B | | | | Deepseek-7B | | | |
|---|---|---|---|---|---|---|---|---|
| Prefix Lens | 0 | 5 | 10 | 50 | 0 | 5 | 10 | 50 |
| Editing Acc. | 98.50% | 84.15% | 80.35% | 77.40% | 99.50% | 90.35% | 84.25% | 85.10% |
| low $\mathcal{A}(i)$ Acc. | - | 94.00% | 91.00% | 90.00% | - | 94.00% | 92.00% | 91.00% |
| top $\mathcal{A}(i)$ Acc. | - | 46.00% | 46.00% | 45.00% | - | 55.00% | 54.00% | 47.00% |

**Empirical Verification of Corollary 1.** We edit 200 CounterFact facts on LLaMA2-7B and Deepseek-7B using MEMIT, and evaluate average editing accuracy with prefix lengths ranging from 5 to 50 tokens. To assess the impact of embedding misalignment, we also compare the 10 items with the highest and lowest misalignment scores under long-prefix conditions.

Table 2 shows that editing performance degrades when edited facts are evaluated with prefixes. For LLaMA2-7B, accuracy falls from 98.5% to 84.15% with a 5-token prefix, and further to 77.40% with a 50-token prefix. A similar trend is observed for Deepseek-7B, where editing accuracy drops by around 15% under 50-token prefixes. The robustness to prefix perturbations varies markedly between items prone to embedding misalignment and those that are not. Consistent with **Corollary 1**, items with lower misalignment scores maintain above-average accuracy, whereas highly misaligned items suffer a sharp decline, with accuracy dropping to an average of 48%.

**Corollary 2.** *Massive editing suffers from degraded performance when multiple samples with the same subject are edited simultaneously. In this case, the reconstruction weight on the target $\beta_{ii}$ decreases while cross-weights $\beta_{ij}$ increase, eventually leading to reconstruction failure of $r_i$.*

Figure 3: Impact of editing same-subject samples. Shaded region indicates shared items.

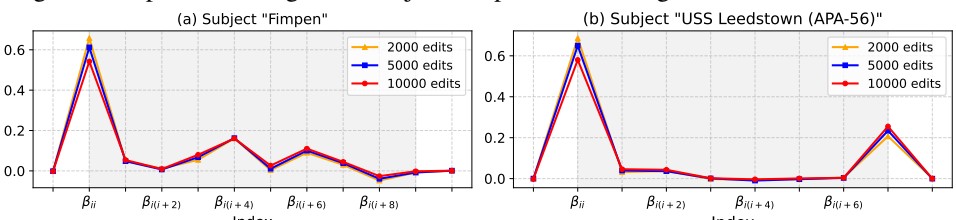

**Empirical Verification of Corollary 2.** Figure 3 shows the reconstruction coefficients for two example subjects under different numbers of edits. Although $\beta_{ii}$ remains the dominant coefficient, its value decreases steadily as the number of co-edited samples increases, while off-diagonal coefficients $\beta_{ij}$ grow accordingly. As a result, $\Delta k_i$ is no longer primarily aligned with $r_i$ but is instead reconstructed as a mixture of other $r_j$, making the recovery of the correct target representation increasingly difficult.

This behavior is consistent with the misalignment measure $\mathcal{A}(i)$, as only $\beta_{ij}$ from the same subject as $i$ take relatively large values, while cross-subject weights remain negligible. Therefore, reconstruction is dominated by the same-subject neighborhood. When $\mathcal{A}(i)$ is small, same-subject embeddings are well aligned across both $k$-space and $r$-space. Thus, using other $r_j$ from the same subject to approximate $r_i$ introduces only limited error. However, when $\mathcal{A}(i)$ is large, misalignment within this neighborhood amplifies the effect of weight redistribution, causing the residual $\|e_i\|$ to grow and ultimately leading to degraded editing performance. We provide details in Appendix C.

These findings underscore the strong connection between embedding misalignment and the effectiveness as well as robustness of massive editing. Motivated by this observation, the following section introduces our approach for aligning key and residual embeddings to enhance the overall performance of massive editing.

## 5 EMBEDDING ALIGNMENT MEMORY OPTIMIZATION

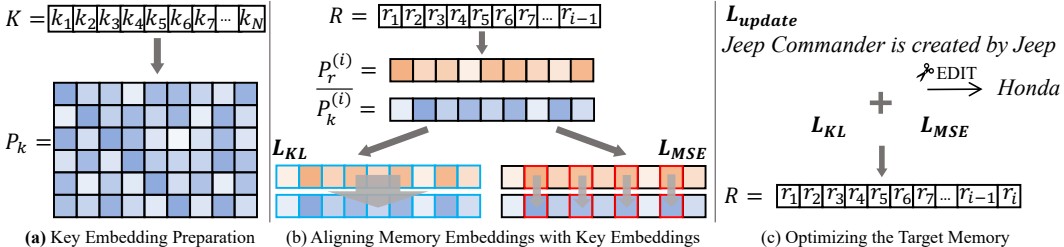

Figure 4: Method Overview of EAMET.

Motivated by these results, we propose *EAMET*, which optimizes memory embeddings to promote alignment with key embeddings across facts. This design enhances the model's ability to edit multiple facts concurrently under practical metrics, while also improving robustness against prefix perturbations and simultaneous edits of same-subject samples. We elaborate on the details below.

**Key Embedding Preparation (Figure 4 (a)).** Before optimization, we extract the key embeddings corresponding to each knowledge item that is scheduled for editing. For a given knowledge item $i$, we calculate the cosine similarity between its key embedding $k_i$ and the key embeddings of all other items. We then collect these similarity values into the set $P_k^{(i)} = \{P_k^{(i,j)} = \cos(k_i, k_j) \mid j \neq i\}$.

**Aligning Memory Embeddings with Key Embeddings (Figure 4 (b)).** For $N$ knowledge items, we separately optimize the target memory embeddings to update factual associations. During the iterative optimization process, we save every optimized residual embedding. When optimizing the target memory for the $i$-th knowledge item, we compute the cosine similarity between $r_i$ and all residual embeddings saved so far, and collect them as $P_r^{(i)} = \{P_r^{(i,j)} \mid j < i\}$. To promote alignment between key and residual embeddings, we compute the KL divergence Chen et al. (2020); He et al. (2020); Sun & Saenko (2016) between $P_r^{(i)}$ and $\bar{P}_k^{(i)}$, where $\bar{P}_k^{(i)} = \{P_k^{(i,j)} \mid j < i\}$ denotes the subset of $P_k^{(i)}$ corresponding to earlier items:

$$L_{\text{KL}}(i) = \text{KL}\left(P_r^{(i)} \,\|\, \bar{P}_k^{(i)}\right). \tag{12}$$

Since KL divergence emphasizes distributional differences, we further strengthen the alignment by selecting the top $M$ cosine similarities $\{P_k^{(i,j)}\}$ from $P_k^{(i)}$, and computing the mean squared error (MSE) loss between the corresponding residual similarities $\{P_r^{(i,j)}\}$:

$$L_{\text{MSE}}(i) = \frac{1}{M} \sum_{j=1}^{M} \left\| P_r^{(i,j)} - P_k^{(i,j)} \right\|^2. \tag{13}$$

**Optimizing the Target Memory (Figure 4 (c)).** Our goal in this step is to compute the residual update vector $r_i$ for each factual association $(s_i, rel_i, o_i)$ such that the model reliably predicts the target object $o_i$ while preserving the alignment between the memory embeddings and the key embeddings. To make the optimization procedure explicit, we describe each component of the objective in

Equation (14). For each fact $i$, let $h_i^L$ denote the hidden state at layer $L$ produced by the templated prompt $tp(s_i, rel_i)$. Following prior work (Meng et al., 2022; 2023), we augment this prompt with a set of $N_{\mathrm{FP}}$ randomly sampled prefixes $\{f_j\}_{j=1}^{N_{\mathrm{FP}}}$, forming inputs $f_j \oplus tp(s_i, rel_i)$. These prefixes encourage the model to learn more generalizable memory representations. We write the forward pass of the model with the edited hidden state as $G_{(h_i^L +=r_i)}$, indicating that the hidden representation at layer $L$ is perturbed by the update vector $r_i$.

Given these definitions, we optimize $r_i$ by minimizing the following loss:

$$r_i = \arg\min_{r_i} \left( \frac{1}{N_{\mathrm{FP}}} \sum_{j=1}^{N_{\mathrm{FP}}} -\log \mathbb{P}_{G(h_i^L +=r_i)}[o_i \,|\, f_j \oplus tp(s_i, rel_i)] + \lambda_{\mathrm{KL}} L_{\mathrm{KL}}(i) + \lambda_{\mathrm{MSE}} L_{\mathrm{MSE}}(i) \right).$$
(14)

Here, the first term encourages the model to predict the correct target object $o_i$ under all sampled prefixes. The losses $L_{\mathrm{KL}}(i)$ and $L_{\mathrm{MSE}}(i)$ ensure that the alignment between the memory embeddings and the key embeddings is preserved, with $\lambda_{\mathrm{KL}}$ and $\lambda_{\mathrm{MSE}}$ controlling their relative importance.

The full optimization procedure is detailed in Appendix E. We justify our design of combining KL loss and MSE loss in Appendix F.5. As the optimization process is iterative, the editing order of knowledge items may influence the performance of EAMET. We further investigate the robustness of EAMET against different editing orders in Table 6.

## 6 EXPERIMENTS

In this section, we empirically focus on evaluating the following research questions (RQs). We first demonstrate the *effectiveness* of EAMET in massive by considering:

- **RQ1.** Can EAMET generate more aligned embeddings for different knowledge items?
- **RQ2.** How does EAMET perform on massive editing tasks compared with baselines for various LLMs? Can it excel under the practical metric?

We then examine the *robustness* of EAMET in two representative scenarios:

- **RQ3.** How does EAMET perform when evaluating edited facts with prefixes?
- **RQ4.** How does EAMET perform when editing multiple facts of the same subject?

### 6.1 EXPERIMENTS SETUP

**Models, Datasets, and Baselines.** We conduct extensive experiments on various LLMs, including LLaMA2-7B (Touvron et al., 2023a), LLaMA2-13B (Touvron et al., 2023a), Falcon-7B (Almazrouei et al., 2023), Qwen-2.5-7B (Yang et al., 2024a), Deepseek-base-7B (Bi et al., 2024), and LLaMA3-8B (Touvron et al., 2023b). We provide additional evaluations on more LLMs in Appendix F.3. We consider a range of ME techniques as baselines: FT (Zhu et al., 2020), MEND (Mitchell et al., 2022a), ROME (Meng et al., 2022), MEMIT (Meng et al., 2023), PMET (Li et al., 2025), and ALPHAEDIT (Fang et al., 2024). We demonstrate their performance on Counter-Fact (Meng et al., 2022), ZsRE (Levy et al., 2017), and Wiki-recent (Zhang et al., 2024). We provide a full description in Appendix D.1.

**Evaluation Metrics.** Following previous work, we evaluate the performance of ME techniques in terms of efficacy (Eff.), generalization (Gen.), specificity (Spe.), and fluency (Flu.) for CounterFact and ZsRE datasets. For Wiki-recent, we additionally evaluate the portability (Zhang et al., 2024) (Por.) of edited models, which represents the ability to address downstream tasks with edited knowledge. We propose to evaluate the editing performance of ME techniques by requiring the edited models to strictly examine whether explicit target objects are retrieved, as demonstrated in Figure 1. The editing efficacy is then defined as:

$$\text{Eff.} = \mathbb{E}_i[o_i = \arg\max_o \mathbb{P}_{f_\theta}(o \,|\, (s_i, rel_i))].$$
(15)

When evaluating efficacy on the CounterFact and Wiki-recent datasets, and generalization on CounterFact, we prepend each prompt with 10 distinct 5-token prefixes. Full details of metrics are provided in Appendix D.2. We also provide the implementation details of EAMET in Appendix D.3.

## 6.2 Alignment of Retrieved Embeddings (RQ1)

***Finding 1.*** **EAMET Promotes More Aligned Embeddings.** We compute the summation of the misalignment score between the residual and key embeddings for 10,000 facts edited by MEMIT, PMET, and EAMET under various LLMs. As shown in Table 3, the residual embeddings generated by EAMET are more aligned with the key embeddings, while those produced by MEMIT and PMET are more likely to cause inconsistency in the key and residual embeddings space.

Table 3: Misalignment score comparison between different methods. Here, "CF" and "ZS" denote the CounterFact and ZsRE datasets, respectively.

| Model | EAMET | | MEMIT | | PMET | |
|---|---|---|---|---|---|---|
| | CF | ZS | CF | ZS | CF | ZS |
| LLaMA2-7B | 377 | 165 | 11506 | 22245 | 11475 | 11477 |
| Qwen-7B | 374 | 180 | 18498 | 23699 | 18471 | 18463 |
| Deepseek-7B | 520 | 161 | 12135 | 23241 | 12155 | 12046 |
| Falcon-7B | 385 | 181 | 8564 | 17589 | 8602 | 8590 |

This observation supports our hypothesis that EAMET encourages more aligned target memory embeddings.

## 6.3 Performance of Massive Editing (RQ2)

Table 4: Performance comparison of different editing methods on six LLMs over the Counterfact, Wiki-recent, and ZsRE benchmarks. We report the average value calculated over five evaluations.

| Model | Method | Counterfact | | | | Wiki-recent | | | | ZsRE | | |
|---|---|---|---|---|---|---|---|---|---|---|---|---|
| | | Eff.↑ | Gen.↑ | Spe.↑ | Flu.↑ | Eff.↑ | Por.↑ | Loc.↑ | Flu.↑ | Eff.↑ | Gen.↑ | Spe.↑ |
| LLaMA2-7B | FT | 0.29 | 0.23 | 77.43 | 490.34 | 7.23 | 41.61 | 36.52 | 491.83 | 5.30 | 4.31 | 14.69 |
| | MEND | 0.23 | 0.31 | **78.55** | 307.26 | 0.00 | 34.67 | 37.46 | 269.52 | 0.00 | 0.00 | 0.50 |
| | ROME | 0.00 | 0.00 | 50.73 | 467.76 | 76.73 | 49.31 | 51.51 | 497.53 | 37.29 | 6.86 | 10.27 |
| | MEMIT | 24.95 | 22.68 | 63.84 | 506.69 | 34.75 | 44.93 | 46.72 | 504.18 | 76.63 | 64.06 | 15.57 |
| | PMET | 74.22 | 46.45 | 72.47 | 507.10 | 81.84 | 51.11 | 53.16 | 497.49 | 77.29 | 71.40 | 16.54 |
| | ALPHAEDIT | 0.51 | 0.53 | 51.14 | 501.63 | 0.07 | 35.34 | 37.48 | **527.83** | 44.26 | 35.83 | 12.65 |
| | **EAMET** | **89.09** | **61.21** | 72.19 | **519.89** | **93.23** | 53.13 | 54.61 | 503.52 | **89.47** | **81.34** | **15.70** |
| Qwen2.5-7B | FT | 16.18 | 14.15 | 56.07 | 527.56 | 21.17 | 51.40 | 51.50 | **515.87** | 14.30 | 13.00 | 39.28 |
| | MEND | 0.01 | 0.06 | 70.73 | 282.92 | 0.00 | 42.55 | 44.37 | 272.90 | 0.00 | 0.00 | 0.09 |
| | ROME | 0.00 | 0.00 | 49.83 | 523.45 | 16.28 | 46.52 | 46.61 | 502.37 | 4.10 | 3.43 | 1.30 |
| | MEMIT | 90.06 | 63.86 | 70.53 | 529.27 | 94.88 | 56.97 | **61.23** | 510.43 | 54.12 | 42.96 | 31.57 |
| | PMET | 65.71 | 52.84 | 63.14 | 518.92 | 82.39 | **58.38** | 57.59 | 511.62 | 53.58 | 46.59 | 36.50 |
| | ALPHAEDIT | 83.15 | 55.70 | 67.16 | 514.07 | 94.16 | 57.17 | 59.45 | 510.32 | 44.52 | 34.98 | 25.52 |
| | **EAMET** | **90.49** | **64.37** | **72.18** | **536.67** | **95.61** | 57.46 | 60.28 | 509.06 | **91.03** | **84.80** | **41.20** |
| LLaMA2-13B | FT | 1.23 | 0.07 | 68.57 | 484.56 | 13.90 | 36.89 | 40.09 | 497.21 | 5.95 | 5.10 | 15.16 |
| | ROME | 4.05 | 1.52 | 50.44 | 525.12 | 11.06 | 38.14 | 39.09 | 447.42 | 5.52 | 5.06 | 2.25 |
| | MEMIT | 47.98 | 34.75 | 71.61 | 517.63 | 94.76 | 51.38 | 50.40 | 507.84 | 69.15 | 51.58 | 15.53 |
| | PMET | 78.60 | 38.76 | **81.15** | 526.82 | 88.66 | 49.69 | 47.58 | 501.61 | 53.27 | 35.73 | 15.76 |
| | ALPHAEDIT | 3.03 | 1.9 | 54.97 | 421.97 | 93.68 | 51.65 | 52.33 | **508.82** | 80.27 | 63.66 | 15.32 |
| | **EAMET** | **92.85** | **60.08** | 77.51 | **530.78** | **95.88** | 52.08 | 53.43 | 504.06 | **87.09** | **74.58** | **15.90** |
| Falcon-7B | FT | 14.70 | 13.54 | 56.34 | 167.18 | 23.94 | 50.46 | 49.69 | 351.18 | 13.64 | 12.68 | 32.28 |
| | ROME | 12.85 | 12.56 | 51.48 | 353.38 | 74.57 | 52.10 | 53.64 | 510.92 | 8.39 | 7.3 | 10.29 |
| | MEMIT | 89.21 | 60.85 | 77.56 | 519.92 | 96.04 | 55.23 | 56.91 | 497.35 | 82.93 | 68.93 | 33.64 |
| | PMET | 77.61 | 57.03 | 70.48 | 517.09 | 58.03 | 54.40 | 54.49 | 500.87 | 69.73 | 60.69 | 35.34 |
| | ALPHAEDIT | 87.62 | 58.32 | 72.43 | 500.35 | 96.22 | 55.47 | 58.02 | 493.56 | 53.78 | 40.83 | 22.60 |
| | **EAMET** | **92.37** | **63.91** | **78.94** | **528.98** | **96.94** | **57.08** | **58.58** | 507.56 | **92.38** | **81.15** | **36.71** |
| Deepseek-7B | FT | 2.61 | 2.49 | **81.43** | 519.35 | 18.85 | 48.90 | 52.78 | 500.86 | 15.00 | 12.28 | 39.14 |
| | ROME | 0.26 | 0.30 | 49.82 | 514.72 | 0.55 | 43.17 | 46.02 | 406.64 | 0.81 | 0.78 | 0.75 |
| | MEMIT | 62.11 | 42.01 | 78.04 | 512.16 | 33.65 | 52.28 | 49.05 | 499.49 | 57.10 | 42.58 | 39.12 |
| | PMET | 74.52 | 43.49 | 79.01 | 514.58 | 86.75 | **57.85** | 59.93 | 500.50 | 76.97 | 69.22 | 38.47 |
| | ALPHAEDIT | 22.51 | 14.00 | 59.92 | 479.52 | 18.53 | 48.33 | 48.74 | 483.38 | 73.41 | 57.09 | 34.87 |
| | **EAMET** | **89.74** | **59.98** | 77.73 | 513.93 | **97.15** | 56.43 | **60.45** | **501.09** | **87.27** | **70.02** | **39.87** |
| LLaMA3-8B | FT | 2.68 | 1.30 | 58.16 | 434.67 | 16.05 | 47.51 | 48.84 | 490.71 | 11.75 | 10.48 | 40.53 |
| | ROME | 51.01 | 33.32 | 64.37 | 491.98 | 82.19 | 54.94 | 57.74 | 518.89 | 7.40 | 6.82 | 27.79 |
| | MEMIT | 93.76 | 61.98 | 77.69 | 526.47 | 92.63 | 55.60 | 58.75 | 527.29 | 78.40 | 71.76 | 39.21 |
| | PMET | 77.71 | 49.41 | 71.43 | 510.82 | 75.81 | 56.89 | 58.26 | 513.84 | 68.52 | 62.72 | 39.35 |
| | ALPHAEDIT | 58.97 | 33.02 | **85.16** | **537.91** | 65.36 | 51.44 | 53.55 | 516.53 | 64.01 | 57.01 | 40.82 |
| | **EAMET** | **93.87** | **63.74** | 79.07 | 533.30 | **94.36** | **57.88** | **59.48** | 528.23 | **85.68** | **81.34** | **42.39** |

We demonstrate the effectiveness of EAMET in massive editing tasks by comparing it with baseline methods across six popular LLMs. Specifically, we simultaneously edit 10,000 factual associations sampled from the CounterFact and ZsRE datasets. For the Wiki-recent dataset, we modify all 1,266 knowledge items. As shown in Table 4, our key findings are as follows:

***Finding 2.*** **EAMET Consistently Achieves Superior Editing Performance Across All Datasets and Model Architectures.** Across all evaluated datasets, EAMET demonstrates the highest levels of editing efficacy and generalization. On the CounterFact dataset, it consistently outperforms

other methods, particularly on base models such as LLaMA2-7B, LLaMA2-13B, and Deepseek-7B. For example, EAMET achieves 89.09% efficacy and 61.21% generalization on LLaMA2-7B, outperforming the second-best method (PMET) by 15% on both metrics. The gap widens further compared to MEMIT, with improvements of 65% in efficacy and 39% in generalization. Even on more advanced models such as Qwen2.5-7B, Falcon-7B, and LLaMA3-8B, EAMET consistently surpasses all baselines. Furthermore, its advantage becomes more pronounced at larger editing scales. As shown in Table 5, when editing 15,000 knowledge items on Qwen2.5-7B, EAMET achieves 83.66% efficacy, demonstrating a 10% improvement over MEMIT. We additionally report the superior performance of EAMET across diverse semantic scenarios in Appendix F.1.

*Finding 3.* **EAMET Preserves the General Abilities of the Edited models.** In addition to achieving state-of-the-art editing performance, EAMET does not impair the base model's fluency or reasoning abilities. Across all datasets, EAMET consistently attains among the highest specificity and fluency scores. Notably, on the Wiki-recent dataset, EAMET achieves the best portability performance on most base models, indicating that the edited models retain their ability to reason about downstream knowledge related to the edited facts. We also evaluate the general abilities of edited models on GLUE (Wang et al., 2018) and find that EAMET yields minimal deviation from pre-edit performance (Appendix F.2).

Table 5: Performance comparison of different editing methods on Qwen2.5-7B, Falcon-7B, and LLaMA3-8B with 15,000 edits from the CounterFact benchmark.

| Model | Method | Counterfact (15000) | | | |
|---|---|---|---|---|---|
| | | Eff.↑ | Gen.↑ | Spe.↑ | Flu.↑ |
| Qwen2.5-7B | MEMIT | 77.46 | 54.34 | 66.23 | 514.81 |
| | **EAMET** | **83.66** | **55.31** | **69.49** | **528.28** |
| Falcon-7B | MEMIT | 84.60 | 56.13 | **75.51** | 513.82 |
| | **EAMET** | **89.55** | **61.00** | 68.44 | **516.91** |
| LLaMA3-8B | MEMIT | 87.58 | 54.76 | 72.65 | 514.07 |
| | **EAMET** | **91.22** | **62.24** | **73.43** | **531.76** |

Table 6: Impact of editing sequence on EAMET's performance on Counterfact and ZsRE datasets.

| Method | Counterfact | | | | ZsRE | | |
|---|---|---|---|---|---|---|---|
| | Eff.↑ | Gen.↑ | Spe.↑ | Flu.↑ | Eff.↑ | Gen.↑ | Spe.↑ |
| EAMET (original sequence) | 89.09 | 61.21 | 72.19 | 519.06 | 89.47 | 81.34 | 15.70 |
| – random shuffle (seed=0) | 88.21 | 60.79 | 71.84 | 519.21 | 87.63 | 77.42 | 15.56 |
| – random shuffle (seed=1) | 89.11 | 60.78 | 72.03 | 518.84 | 86.99 | 76.08 | 15.58 |
| – random shuffle (seed=2) | 88.91 | 59.38 | 72.34 | 518.23 | 87.56 | 77.47 | 15.59 |

As EAMET preserves previously optimized residual embeddings when updating new knowledge items, the editing sequence could potentially affect its performance. To assess this, we examine EAMET's robustness under different editing orders on the Counterfact and ZsRE datasets. In Counterfact, all knowledge items have distinct subjects, whereas in ZsRE some items share the same subject and are adjacent in the original order. We therefore randomly shuffle the order of 10,000 items three times and report the average performance, alongside the original sequence as a reference.

*Finding 4.* **EAMET is Robust to Editing Sequence.** As shown in Table 6, EAMET's performance remains stable across editing orders. On Counterfact, random shuffles produce only negligible variations in efficacy, generalization, and specificity. On ZsRE, editing efficacy shows a slight decline of about 2%, likely due to the neighborhood structure of items sharing the same subject in the original sequence. Overall, these results suggest that EAMET is largely insensitive to editing order, demonstrating strong robustness to sequence variations.

## 6.4 ROBUSTNESS AGAINST LONG PREFIXES (RQ3)

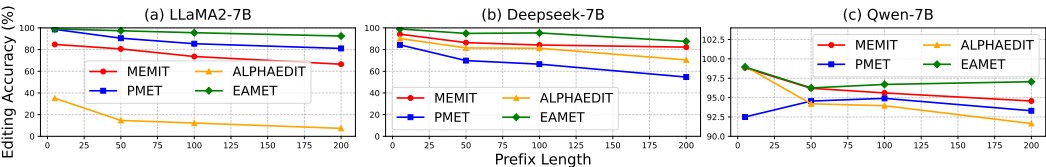

Figure 5: Editing performance of different methods across varying prefix lengths.

We evaluate the robustness of editing methods when edited facts are preceded by varying numbers of tokens. Specifically, we modify 200 facts from the CounterFact dataset in LLaMA2-7B, Deepseek-7B, and Qwen2.5-7B. During evaluation, we prepend prefixes of 5, 50, 100, and 200 tokens to them.

***Finding 5.*** **EAMET Remains Effective When Edits Are Preceded by Long Prefixes.** In Figure 5, EAMET achieves the highest editing efficacy across all models, with at most a 7% drop at 200-token prefixes. In contrast, MEMIT suffers a much larger decline, from 84.75% to 66.50% on LLaMA2-7B and from 94.2% to 82.25% on DeepSeek-7B. Notably, all methods demonstrate strong robustness on Qwen2.5-7B, consistent with our earlier observation that Qwen2.5-7B is more suitable for robust batch editing. Nevertheless, EAMET exhibits the smallest efficacy drop (only 1.9%) when the prefix increases to 200 tokens, which is half that of the second-best method (MEMIT).

## 6.5 ROBUSTNESS UNDER MULTIPLE EDITS OF THE SAME SUBJECT (RQ4)

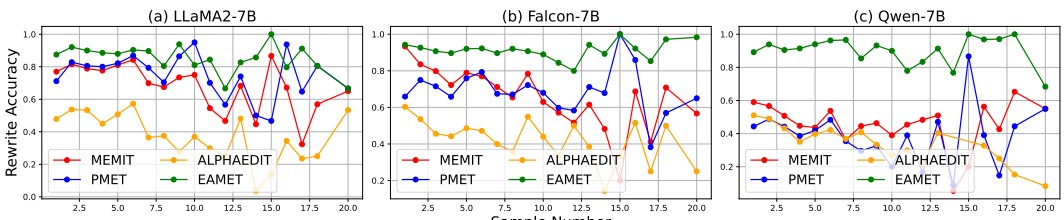

Figure 6: Editing performance of different methods across varying numbers of facts per subject.

We evaluate the robustness of editing methods when multiple facts concerning the same subject are edited simultaneously. Specifically, we simultaneously edit 10,000 facts from ZsRE dataset, and only evaluate samples whose subject is associated with multiple facts. We group subjects according to the number of associated samples and examine how rewrite accuracy varies with this number. Experiments are conducted on LLaMA2-7B, DeepSeek-7B, and Qwen2.5-7B.

***Finding 6.*** **EAMET Remains Effective When Multiple Facts of the Same Subject Are Edited Simultaneously.** Figure 6 shows that EAMET consistently achieves the highest editing efficacy across nearly all settings. Its performance remains stable when editing multiple samples associated with the same subject. In contrast, other methods exhibit a clear decline in efficacy as the number of facts per subject increases, which ultimately results in degraded performance on the overall massive editing task.

## 7 CONCLUSION

In this paper, we propose EAMET, a novel model editing method that enables stronger and more robust massive editing across various models and datasets. We first identify that the failures of existing methods in both effectiveness and robustness of massive editing stem from misalignment between the space of key and residual embeddings. EAMET addresses this issue by progressively aligning the key and residual embedding space when optimizing target memory for each fact. The aligned embeddings increase both the capacity and robustness of massive editing. Extensive experiments on multiple base LLMs, including LLaMA2, LLaMA3, Deepseek, and Qwen, demonstrate that EAMET significantly outperforms existing methods in editing performance and robustness.

## ACKNOWLEDGMENTS

The HKUST authors were supported in part by a RGC GRF grant under the contract 16214723, research fund provided by CMHK and ZTE, a Cohere Labs Catalyst Grant Program under the project "Repairing the Multilingual Tool Calling Gap in Agentic LLMs" and a HKUST Bridge The Gap fund BGF.001.2025. We are grateful to the anonymous reviewers for their valuable comments. We thank HKUST Fok Ying Tung Research Institute and National Supercomputing Center in Guangzhou Nansha Sub-center for computational resources.

## ETHICS STATEMENT

This work does not raise any specific ethical concerns. Its primary goal is to advance research on model editing and to offer a new perspective on this problem.

## REPRODUCIBILITY STATEMENT

All experiments in this paper are reproducible. The code and datasets are publicly available at the GitHub repository: `https://github.com/ybdai7/EAMET-massive-editing`.

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

## A  LLM USAGE STATEMENT

We used LLMs solely to assist in drafting and polishing the writing of this paper, without any other purposes.

## B  PROOF OF THEOREM 1

**Notation Setup.** Let $K = [k_1, \ldots, k_N] \in \mathbb{R}^{d \times N}$ and $R = [r_1, \ldots, r_N] \in \mathbb{R}^{d \times N}$, and define

$$M := \sum_{i=1}^{N} k_i k_i^\top = KK^\top, \qquad N := \sum_{i=1}^{N} r_i k_i^\top = RK^\top. \tag{16}$$

Assume that $M$ is (pseudo-)invertible and define the closed-form solution $\Delta = NM^+ = RK^\top(KK^\top)^+$, so that $\Delta \sum_i k_i k_i^\top = \sum_i r_i k_i^\top$.

**Derivation of the Column Expansion $\Delta k_i = \sum_j \beta_{ij} r_j$.** By definition,

$$\Delta = \sum_{j=1}^{N} r_j k_j^\top M^+ = RK^\top M^+. \tag{17}$$

Applying $\Delta$ to a column $k_i$ gives

$$\Delta k_i = \sum_{j=1}^{N} r_j k_j^\top M^+ k_i. \tag{18}$$

Setting $\beta_{ij} := k_j^\top M^+ k_i$, we immediately obtain

$$\Delta k_i = \sum_{j=1}^{N} \beta_{ij} r_j. \tag{19}$$

This formula provides an explicit linear combination of residual embeddings $r_j$ that reconstructs $\Delta k_i$, with coefficients $\beta_{ij}$ determined by the key embeddings and the pseudo-inverse of $M$.

**Reconstruction Residual and Neighborhood Decomposition.** For each knowledge item $i$, define the reconstruction residual $\mathrm{e}_i := \Delta k_i - r_i$. Suppose that $r_i$ can be approximately reconstructed from its neighbors with nonnegative weights $q_{ij}$ for $j \neq i$, i.e.,

$$r_i = \sum_{j \neq i} q_{ij} r_j + \varepsilon_i, \qquad q_{ij} \geq 0, \ \sum_{j \neq i} q_{ij} = 1, \tag{20}$$

where $\varepsilon_i$ denotes the residual error. Substituting this decomposition into $\mathrm{e}_i$ gives

$$\mathrm{e}_i = \sum_{j \neq i} (\beta_{ij} - q_{ij}) \, r_j + \beta_{ii} r_i - \varepsilon_i. \tag{21}$$

**Bounding the Reconstruction Residual.** Taking norms and applying the triangle inequality yields

$$\|\mathrm{e}_i\| \leq \sum_{j \neq i} |\beta_{ij} - q_{ij}| \, \|r_j\| + |\beta_{ii}| \, \|r_i\| + \|\varepsilon_i\|. \tag{22}$$

To relate the first term to embedding alignment, we construct a probability vector $p_i$ from the positive parts of the coefficients $\beta_{ij}$ (for $j \neq i$):

$$s_{ij} := \max\{\beta_{ij}, 0\}, \qquad S_i := \sum_{j \neq i} s_{ij}, \qquad p_{ij} := \frac{s_{ij}}{S_i}. \tag{23}$$

Defining $C_i := \sum_{j \neq i} \|r_j\|$, one can show that

$$\sum_{j \neq i} |\beta_{ij} - q_{ij}| \, \|r_j\| \leq C_i \, \mathrm{TV}(p_i, q_i), \tag{24}$$

up to negligible contributions from negative $\beta_{ij}$ that can be absorbed into $C_i$. Here, $\mathrm{TV}(p_i, q_i)$ is the *total variation (TV) distance* between two discrete distributions $p_i$ and $q_i$:

$$\mathrm{TV}(p, q) := \tfrac{1}{2} \sum_j |p_j - q_j|. \tag{25}$$

Finally, applying Pinsker's inequality $\mathrm{TV}(p_i, q_i) \le \sqrt{\tfrac{1}{2} \mathrm{KL}(q_i \| p_i)}$ (Cover & Thomas, 1999) gives

$$\|\mathrm{e}_i\| \le C_i \sqrt{\frac{1}{2} \mathrm{KL}(q_i \| p_i)} + |\beta_{ii}| \, \|r_i\| + \|\varepsilon_i\|. \tag{26}$$

Identifying $q_i = P_r^{(i)}$ and $p_i = P_k^{(i)}$ with the kernel-normalized neighborhood distributions from Definition 1 yields the embedding-alignment bound stated in the main text:

$$\|\mathrm{e}_i\| \le C_i \sqrt{\frac{1}{2} \mathcal{A}(i)} + |\beta_{ii}| \, \|r_i\| + \|\varepsilon_i\|. \tag{27}$$

If $M$ is singular, replace $M^+$ with the Moore–Penrose pseudoinverse. The contributions from negative $\beta_{ij}$ or scaling factors can usually be absorbed into $C_i$. In the ideal case of perfect neighborhood alignment $\mathcal{A}(i) = 0$, negligible self-weight $\beta_{ii} = 0$, and vanishing residual $\varepsilon_i = 0$, we recover $\mathrm{e}_i = 0$.

## C  DETAILED ANALYSIS OF COROLLARY 2

We provide a detailed theoretical analysis that develops Corollary 2.

**Residual Decomposition.** Recall that the reconstruction residual can be written as

$$e_i = \Delta k_i - r_i = \sum_{j=1}^{N} \beta_{ij} r_j - r_i, \qquad \beta_{ij} := k_j^\top \Big( \sum_{\ell=1}^{N} k_\ell k_\ell^\top \Big)^{-1} k_i. \tag{28}$$

Partition the index set into the subject cluster $\mathcal{S}$ (samples sharing the subject with $i$) and the remainder $\mathcal{T}$. Then

$$\Delta k_i = \beta_{ii} r_i + \sum_{j \in \mathcal{S}, j \neq i} \beta_{ij} r_j + \sum_{j \in \mathcal{T}} \beta_{ij} r_j. \tag{29}$$

Empirically and theoretically, only coefficients $\beta_{ij}$ for $j \in \mathcal{S}$ become significant, while cross-subject coefficients remain negligible since embeddings from different subjects are nearly orthogonal in key space and thus contribute little to the reconstruction. Hence reconstruction is dominated by the same-subject neighborhood.

**Effect of Adding Same-Subject Samples.** Let $K = \sum_\ell k_\ell k_\ell^\top$ denote the Gram matrix. Suppose we add one additional key $k_j$ (with $j \in \mathcal{S}, j \neq i$). By the Woodbury identity(Horn & Johnson, 1985),

$$(K + k_j k_j^\top)^{-1} = K^{-1} - \frac{K^{-1} k_j k_j^\top K^{-1}}{1 + k_j^\top K^{-1} k_j}. \tag{30}$$

Consequently, the updated self-weight becomes

$$\beta_{ii}^{\mathrm{new}} = k_i^\top (K + k_j k_j^\top)^{-1} k_i = \beta_{ii} - \frac{(k_i^\top K^{-1} k_j)^2}{1 + k_j^\top K^{-1} k_j}. \tag{31}$$

Thus $\beta_{ii}$ monotonically decreases as more same-subject vectors are included. The lost weight is redistributed into off-diagonal terms $\beta_{ij}$, consistent with our empirical observation in Figure 3.

**Connection to Alignment.** Using the decomposition in equation 29, the residual can be expressed as

$$e_i = \sum_{j \in \mathcal{S}} \beta_{ij} (r_j - r_i) + \sum_{j \in \mathcal{T}} \beta_{ij} r_j. \tag{32}$$

Applying the triangle inequality yields

$$\|e_i\| \leq \sum_{j \in \mathcal{S}} |\beta_{ij}| \, \|r_j - r_i\| + \sum_{j \in \mathcal{T}} |\beta_{ij}| \, \|r_j\|. \tag{33}$$

The first term depends on the dispersion of responses within the same subject. This dispersion is controlled by the alignment measure $\mathcal{A}(i)$: when $\mathcal{A}(i)$ is small, the responses $\{r_j : j \in \mathcal{S}\}$ are tightly clustered around $r_i$, so even a redistribution of weight from $\beta_{ii}$ to other $\beta_{ij}$ produces only minor error. Conversely, when $\mathcal{A}(i)$ is large, intra-subject responses differ substantially, and the redistributed weights amplify reconstruction error.

Combining equation 31 and equation 33, we conclude that co-editing additional same-subject samples (i) monotonically decreases $\beta_{ii}$, (ii) redistributes weight into off-diagonal $\beta_{ij}$, and (iii) yields residuals bounded by the intra-subject alignment $\mathcal{A}(i)$. Therefore, massive editing performance crucially depends on the degree of alignment within the subject cluster.

## D    DETAILED EXPERIMENT SETUP

In the following, we provide detailed experimental configurations, including the description of the datasets, introduction of baselines, explanation of evaluation metrics, and implementation details.

### D.1    DATASETS AND BASELINES

We evaluate the performance of model editing techniques using the following datasets:

- **CounterFact** (Meng et al., 2022) is a benchmark for evaluating factual knowledge localization and editing in LLMs. It contains 21,917 entries that describe the named entities along with their counterfactual variations. Model editing techniques could be evaluated in terms of editing efficacy, generalization, and locality. The benchmark also contains generation prompts to test the model's generation ability after editing.

- **ZsRE** (Levy et al., 2017) is a question-answering (QA) benchmark designed to evaluate zero-shot relation extraction capabilities of language models. Entries in the benchmark consist of a subject entity along with an answer as the editing target. The benchmark also includes paraphrased questions for testing generalization ability and irrelevant questions for evaluating the locality of editing techniques.

- **Wiki-recent** (Zhang et al., 2024) contains 1,266 entries of triplets that have been added into WIKIDATA after July 2022. The benchmark enables insertion for models that were trained prior to the introduction of these facts. This simulates the cases of editing outdated models with newly introduced facts. Model editing techniques are evaluated in terms of editing efficacy, portability, and locality. Here, portability emphasizes whether the edited model could reason about the downstream effects of facts when they are inserted into the model.

We proceed to introduce baseline methods evaluated in the paper. For all baseline methods, we use the official implementation provided by the authors.

- **MEND** (Mitchell et al., 2022a) requires extra parameters for efficiently editing pretrained LLMs. It introduces a set of small auxiliary networks that transform standard fine-tuning gradients into low-rank updates, enabling fast and localized edits without retraining the entire model. This approach offers a scalable solution for post-hoc model editing, avoiding the overfitting issue of traditional fine-tuning methods.

- **ROME** (Meng et al., 2022) performs factual knowledge editing by directly modifying the feed-forward weights in specific layers of LLMs. It first identifies that factual knowledge is primarily stored in mid-layer feed-forward modules, thereby demonstrating the feasibility of editing model parameters to update internal knowledge. The method then updates these weights to encode specific factual associations. ROME achieves precise insertion of new facts with minimal interference to unrelated knowledge. When evaluating using ROME, we edit all facts sequentially with batch size 1, as it does not support batch editing.

- **MEMIT** (Meng et al., 2023) is designed to efficiently update LLMs with thousands of factual associations simultaneously. Building upon ROME, MEMIT employs a least-squares optimization over multiple key-value memory components, ensuring high specificity and minimal interference with unrelated knowledge. It further distributes the updates across multiple layers, which helps reduce the impact on the model's general capabilities.

- **PMET** (Li et al., 2025) is a method designed to enhance the precision of knowledge updates in large language models. Unlike prior approaches that treat transformer layer (TL) hidden states as direct inputs of the feed-forward network (FFN), PMET recognizes that these hidden states also encompass information from multi-head self-attention (MHSA) and residual connections. PMET proceeds to simultaneously optimize MHSA and FFN hidden states and use the optimized TC hidden states of FFN to precisely update FFN weights. This approach enables more accurate and efficient model editing, preserving the integrity of the model's existing knowledge while incorporating new information.

- **ALPHAEDIT** (Fang et al., 2024) preserves knowledge in LLMs during sequential updates by projecting updates onto the null space of the preserved knowledge. This could ensure that new modifications do not interfere with previously stored information. This approach maintains the integrity of the model's existing knowledge while enabling precise edits.

## D.2 METRICS

We now introduce the metrics used for CounterFact, Wiki-recent and ZsRE respectively.

### D.2.1 COUNTERFACT METRICS

Given an LLM $f_\theta$, a knowledge fact tuple (subject $s_i$, relation $r_i$), a target output $o_i$ and the original output $o_i^c$, we define the following metrics:

- **Editing Efficacy**: Unlike previous works that evaluate the portion of cases where $o_i$ is more probable than $o_i^c$, we directly compute the average top-1 accuracy of edited samples.

$$\mathbb{E}_i[o_i = \arg\max_o \mathbb{P}_{f_\theta}(o \mid (s_i, r_i))] \tag{34}$$

- **Generalization**: Average top-1 accuracy of the edited model on rephrased statements $N((s_i, r_i))$ of the original knowledge fact. Rephrased statements share the same semantic meaning with the original statements.

$$\mathbb{E}_i[o_i = \arg\max_o \mathbb{P}_{f_\theta}(o \mid N((s_i, r_i)))] \tag{35}$$

- **Specificity**: The portion of cases where $o_i^c$ is more probable than $o_i$ with neighboring statements $O((s_i, r_i))$. Neighboring statements are constructed using prompts which share distinct but semantically related subjects with the original knowledge fact.

$$\mathbb{E}_i[\mathbb{P}_{f_\theta}(o_i^c \mid O((s_i, r_i))) > \mathbb{P}_{f_\theta}(o_i \mid O((s_i, r_i)))] \tag{36}$$

- **Fluency**: Fluency score measures the quality of the generated text. It scores low if the generated text contains excessive repetition.

$$-\frac{2}{3}\sum_k g_2(k)\log_2 g_2(k) + \frac{4}{3}\sum_k g_3(k)\log_2 g_3(k) \tag{37}$$

where $g_2(k)$ and $g_3(k)$ are the probabilities of bigram and trigram $k$ respectively.

### D.2.2 WIKI-RECENT METRICS

Given a LLM $f_\theta$, a knowledge fact tuple $(s_i, r_i)$, a target output $o_i$ and the original output $o_i^c$, we define the following metrics:

- **Editing Efficacy**: Average top-1 accuracy of edited samples.

$$\mathbb{E}_i[o_i = \arg\max_o \mathbb{P}_{f_\theta}(o \mid (s_i, r_i))] \tag{38}$$

- **Portability**: Average top-1 accuracy of the edited model on portability prompts $P((s_i, r_i))$ of the original knowledge fact. Portability prompts contain three parts: alias prompts, compositionality and reasoning prompts, and logical generation prompts. Specifically, alias prompts are constructed by replacing the subject $s_i$ with an alias or synonym. Compositionality and reasoning prompts require the post-edit model to conduct reasoning about the changed fact. Logical generation prompts are changes that are semantically related to the modified fact and expected to change by the edit.

$$\mathbb{E}_i[o_i = \arg\max_o \mathbb{P}_{f_\theta}(o \mid P((s_i, r_i)))] \tag{39}$$

- **Locality**: Average top-1 accuracy of the edited model on neighboring prompts $O((s_i, r_i))$ of the original knowledge fact.

$$\mathbb{E}_i[o_i = \arg\max_o \mathbb{P}_{f_\theta}(o \mid O((s_i, r_i)))] \tag{40}$$

- **Fluency**: Fluency score measures the quality of the generated text. It scores low if the generated text contains excessive repetition.

$$-\frac{2}{3} \sum_k g_2(k) \log_2 g_2(k) + \frac{4}{3} \sum_k g_3(k) \log_2 g_3(k) \tag{41}$$

where $g_2(k)$ and $g_3(k)$ are the probabilities of bigram and trigram $k$ respectively.

### D.2.3  ZsRE Metrics

Given a LLM $f_\theta$, a knowledge fact tuple $(s_i, r_i)$, a target output $o_i$ and the original output $o_i^c$, we define the following metrics:

- **Editing Efficacy**: Average top-1 accuracy of edited samples.

$$\mathbb{E}_i[o_i = \arg\max_o \mathbb{P}_{f_\theta}(o \mid (s_i, r_i))] \tag{42}$$

- **Generalization**: Average top-1 accuracy of the edited model on generalization prompts $N((s_i, r_i))$ of the original knowledge fact.

$$\mathbb{E}_i[o_i = \arg\max_o \mathbb{P}_{f_\theta}(o \mid N((s_i, r_i)))] \tag{43}$$

- **Locality**: Average top-1 accuracy of the edited model on neighboring prompts $O((s_i, r_i))$ of the original knowledge fact.

$$\mathbb{E}_i[o_i = \arg\max_o \mathbb{P}_{f_\theta}(o \mid O((s_i, r_i)))] \tag{44}$$

### D.2.4  Examples of Evaluation Prompts with Prefixes

We provide examples of evaluation prompts with prefixes in the following. For the evaluated prompt "The mother tongue of Danielle Darrieux is French", corresponding evaluated prompts with ten distinct 5-token prefixes are:

---

**Example Prompts with Prefixes**

- The doctor's office was too. The mother tongue of Danielle Darrieux is French.
- Therefore, the average speed of. The mother tongue of Danielle Darrieux is French.
- You can find many examples of. The mother tongue of Danielle Darrieux is French.
- However, the government's new. The mother tongue of Danielle Darrieux is French.
- And so, as the night. The mother tongue of Danielle Darrieux is French.
- While the world of sports and. The mother tongue of Danielle Darrieux is French.
- To make a cake, you. The mother tongue of Danielle Darrieux is French.
- Nevertheless, the overall sentiment of. The mother tongue of Danielle Darrieux is French.
- Never tried it before. The mother tongue of Danielle Darrieux is French.
- He realized that the people of. The mother tongue of Danielle Darrieux is French.

---

For model evaluation, we generate prefixes by prompting the unedited model with the following initial words: "The", "Therefore", "You", "However", "And", "While", "To", "Nevertheless", "Never", and "He". These initial words are used to generate diverse 5-token prefixes, which are then prepended to each edited fact during the evaluation process. This approach ensures a comprehensive assessment of the model's performance across different linguistic contexts.

### D.3 IMPLEMENTATION DETAILS

We implement all experiments on a single NVIDIA H800 GPU with 80GB memory. During optimization, we iterate for 25 steps with 0.5 learning rate. We set $M = 50$ for balancing the fine-grained alignment and optimization efficiency. The details of our implementation across different models are outlined as follows:

- **LLAMA2-7B**: We modify layers [3, 4] for editing factual knowledge. The hyperparameters $\lambda$s are set to [4000, 4000] respectively for two layers. We set $\lambda_{KL} = 2$ and $\lambda_{MSE} = 8$.

- **Qwen-2.5-7B**: We modify layers [3, 4] for editing factual knowledge. The hyperparameters $\lambda$s are set to [500, 500] respectively for two layers. We set $\lambda_{KL} = 1.5$ and $\lambda_{MSE} = 8$.

- **LLaMA2-13B**: We modify layers [3, 4] for editing factual knowledge. The hyperparameters $\lambda$s are set to [4000, 4000] respectively for two layers. We set $\lambda_{KL} = 1.5$ and $\lambda_{MSE} = 8$.

- **Falcon-7B**: We modify layers [3, 4] for editing factual knowledge. The hyperparameters $\lambda$s are set to [1000, 1000] respectively for two layers. We set $\lambda_{KL} = 2$ and $\lambda_{MSE} = 8$.

- **Deepseek-base-7B**: We modify layers [3, 4] for editing factual knowledge. The hyperparameters $\lambda$s are set to [4000, 4000] respectively for two layers. We set $\lambda_{KL} = 4.5$ and $\lambda_{MSE} = 8$.

- **LLaMA3-8B**: We modify layers [3, 4] for editing factual knowledge. The hyperparameters $\lambda$s are set to [1000, 1000] respectively for two layers. We set $\lambda_{KL} = 2$ and $\lambda_{MSE} = 8$.

We justify our choice of editing layers by comparing against the common settings used in prior work (Meng et al., 2023; Li et al., 2025; Meng et al., 2022). On LLaMA2-7B, we evaluate EAMET with different layer combinations when editing 10,000 facts from CounterFact and ZsRE. As shown in Table 7, EAMET achieves higher efficacy and generalization with layers [3, 4] compared to [4, 5, 6, 7, 8].

Table 7: Performance comparison of EAMET on LLAMA2-7B with different layer selections.

| Layers | Counterfact | | | | ZsRE | | |
|---|---|---|---|---|---|---|---|
| | Eff.↑ | Gen.↑ | Spe.↑ | Flu.↑ | Eff.↑ | Gen.↑ | Spe.↑ |
| 3, 4 | 89.09 | 61.21 | 72.19 | 519.23 | 89.47 | 81.34 | 15.70 |
| 4, 5, 6, 7, 8 | 77.58 | 36.83 | 73.43 | 516.63 | 87.14 | 76.91 | 15.92 |

This effect arises because, as the edited layer becomes deeper, the similarity between key embeddings of different knowledge items increases. As shown in Figure 7, the average similarity across layers of LLaMA2-7B grows with layer depth. When the last edited layer is 8, the average similarity is nearly twice that of layer 4. This growth makes it more difficult to align the memory embedding space with the key embedding space, since KL divergence primarily captures distributional differences and we only apply MSE loss to the top-$M$ cosine similarities. Applying MSE to all cosine similarities may lead to vanishing gradients. As the number of similarities grows, the strongest ones become diluted, which slows convergence and hinders optimization. Such misaligned memory embeddings can substantially degrade both the effectiveness and robustness of massive editing.

## E ALGORITHMIC DESCRIPTION OF EAMET

In this section, we present a detailed description of the EAMET algorithm in Algorithm 1. The procedure consists of three main stages: (1) key embedding preparation, (2) aligning memory embeddings with key embeddings, and (3) distributing MLP updates across candidate layers.

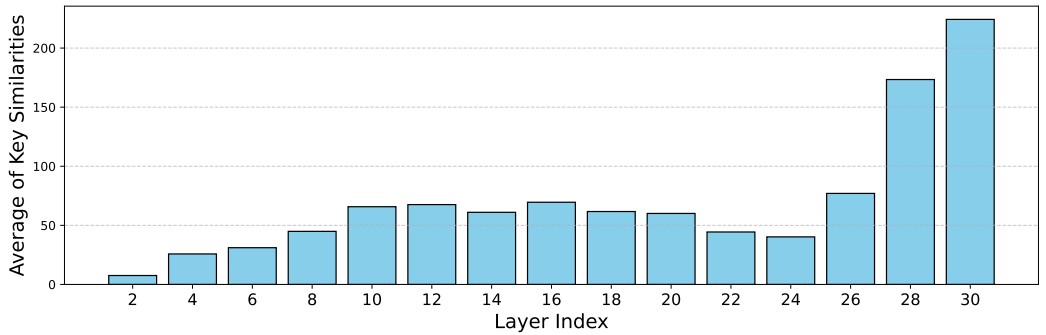

Figure 7: Average of key similarities across different layers of LLaMA2-7B when editing 500 knowledge items.

---

**Algorithm 1:** The EAMET Algorithm

---

1 **Data**: Requested edits $\mathcal{E} = \{(s_i, rel_i, o_i)\}$, generator $G$, layers to edit $\mathcal{S}$, covariances $C^l$
2 **Result**: Modified generator containing edits from $\mathcal{E}$
3
4 $K \leftarrow []$
5 $L \leftarrow$ final layer of candidate layers $\mathcal{R}$
6 **for** $s_i, rel_i, o_i \in \mathcal{E}$ **do**
7 $\quad$ $k_i^L \leftarrow k_i^L = \frac{1}{N_{FP}} \sum_{j=1}^{N_{FP}} k(f_j \oplus s_i)$
8 $\quad$ $K \leftarrow K \cup \{k_i^l\}$

9 $P_k \leftarrow \{P_k^{(i,j)} = \cos(k_i, k_j) \mid j \neq i, k_i, k_j \in K\}$
10 $R \leftarrow []$
11 **for** $s_i, rel_i, o_i \in \mathcal{E}$ **do**
12 $\quad$ $r_i \leftarrow h_i^L$ $\qquad\qquad$ // Initialize $r_i$ as the original hidden state
13 $\quad$ $P_r^{(i)} \leftarrow \{P_r^{(i,j)} \mid j < i, r_j \in R\}$
14 $\quad$ $\bar{P}_k^{(i)} \leftarrow \{P_k^{(i,j)} \mid j < i, k_j \in K\}$
15 $\quad$ $L_{\text{KL}}(i) = \text{KL}\left(P_r^{(i)} \parallel \bar{P}_k^{(i)}\right)$
16 $\quad$ $I_K \leftarrow$ indices of the top $M$ largest elements in $\bar{P}_k^{(i)}$
17 $\quad$ $L_{\text{MSE}}(i) = \frac{1}{M} \sum_{j \in I_K} \left\| P_r^{(i,j)} - P_k^{(i,j)} \right\|^2$
18 $\quad$ $r_i \leftarrow \arg\min_{r_i} \frac{1}{N_{FP}} \sum_{j=1}^{N_{FP}} -\log \mathbb{P}_{G_{(h_i^L += r_i)}}[o_i \mid$
    $\quad\quad f_j \oplus tp(s_i, r_i)] + \lambda_{KL} L_{\text{KL}}(i) + \lambda_{MSE} L_{\text{MSE}}(i)$
19 $\quad$ $z_i \leftarrow h_i^L + r_i$
20 $\quad$ $R \leftarrow R \cup \{r_i\}$

21 **for** $l \in \mathcal{R}$ **do**
22 $\quad$ $h_i^l \leftarrow h_i^{l-1} + a_i^l + m_i^l$
23 $\quad$ **for** $s_i, rel_i, o_i \in \mathcal{E}$ **do**
24 $\quad\quad$ $k_i^l \leftarrow k_i^l = \frac{1}{N_{FP}} \sum_{j=1}^{N_{FP}} k(f_j \oplus s_i)$
25 $\quad\quad$ $r_i^l \leftarrow \frac{z_i - h_i^L}{L - l + 1}$ $\qquad$ // Distribute over remaining layers
26 $\quad$ $K^l \leftarrow [k_1^l, \ldots, k_{N_t}^l]$
27 $\quad$ $R^l \leftarrow [r_1^l, \ldots, r_{N_t}^l]$
28 $\quad$ $\Delta \leftarrow R^l K^{l^T} (C^l + K^l K^{l^T})^{-1}$
29 $\quad$ $W^l \leftarrow W^l + \Delta$ $\qquad$ // Update layer $l$ MLP weights in model

---

**Key Embedding Preparation**. Following prior work (Meng et al., 2022; 2023; Li et al., 2025), we evenly distribute MLP updates across the critical target layers $\mathcal{R}$. We denote by $L$ the final candidate layer where new memories are fully represented (Line 5). Before optimizing residual embeddings, we first compute the key embeddings for each target edit (Line 7). To improve generalization, each subject is augmented with $N_{FP}-1$ random prefixes of fixed length $f_i$. All resulting key embeddings are aggregated into a matrix $K$ (Line 8). We then compute pairwise cosine similarities among key embeddings to obtain $P_k$ (Line 9).

**Aligning Memory Embeddings with Key Embeddings**. For each target edit, we initialize the residual embedding with the original hidden state (Line 12), since it naturally corresponds to the associated key embedding and thus provides a good starting point. We compute cosine similarities among residual embeddings to form $P_r^{(i)}$ (Line 13), and extract the corresponding key embedding structure $\bar{P}_k^{(i)}$ (Line 14). Alignment is achieved by minimizing the KL divergence between $P_r^{(i)}$ and $\bar{P}_k^{(i)}$ (Line 15). To further refine alignment, we select the indices $I_K$ corresponding to the top $M$ similarities in $P_k^{(i)}$ (Line 16) and minimize the MSE loss between $P_r^{(i)}$ and $P_k^{(i)}$ restricted to these indices (Line 17). The optimized residual embedding $r_i$ is obtained by minimizing this combined objective (Line 18) and stored in the set $R$ (Line 19).

**Distributing MLP Updates Across Candidate Layers**. We update MLP modules sequentially across layers $l \in \mathcal{R}$, as earlier edits affect subsequent representations. For each candidate layer, we compute key embeddings $k_i^l$ for all edits (Line 24) and residual embeddings $r_i^l$, distributing them proportionally across layers (Line 25). These embeddings are then aggregated into $K^l$ and $R^l$ (Lines 26-27) to update the MLP weights with $\Delta$ (Lines 28-29).

# F   ADDITIONAL EXPERIMENTAL RESULTS

In this section, we present additional experiments and findings to further validate the effectiveness of EAMET. We begin by evaluating editing performance across different semantic categories. Next, we assess its impact on the model's general capabilities using the GLUE benchmarks. We then report results on two additional LLMs, Gemma-7B (Team et al., 2024) and Phi-1.5 (Li et al., 2023). We also examine how the order of edits affects EAMET's performance. Furthermore, we explore its integration with sequential editing, showing that embedding alignment enables larger batch sizes per step and thus reduces the number of steps needed to edit the same set of knowledge items. Finally, we provide an ablation study on combining KL loss and MSE loss, along with a comprehensive analysis of the hyperparameters $\lambda_{KL}$, $\lambda_{MSE}$, and $M$.

## F.1   EDITING PERFORMANCE INVOLVING DIFFERENT SEMANTICS

***Additional Finding 1.*** **EAMET Achieves Superior Editing Performance Across Different Semantics.** We extract samples with specific relation types from the CounterFact dataset to evaluate the performance of different editing methods across semantic categories. As shown in Figure 8, EAMET consistently achieves the highest editing efficacy and generalization on both LLaMA2-7B and Qwen2.5 for most semantic types. On LLaMA2-7B, EAMET outperforms the second-best method (PMET) by approximately 10% in efficacy and 20% in generalization. In terms of editing specificity, EAMET performs better on Qwen2.5 than on LLaMA2. On LLaMA2, PMET surpasses EAMET by an average of 5%, whereas on Qwen2.5, EAMET achieves the highest specificity on 6 out of 8 relation types.

We observe an interesting phenomenon on the *twin-city* relation of LLaMA2-7B: EAMET achieves $2\times$ higher efficacy and $4\times$ higher generalization compared to PMET, while MEMIT nearly fails on this relation, yielding efficacy and generalization scores close to 0%. This occurs because facts involving the twin-city relation are typically expressed in forms such as The twin city of subject or What is the twin city of subject?. The key embeddings, which are extracted from the last subject token, are therefore highly similar across facts due to the shared prefixes in these templates. As a result, reconstructing each individual update $\Delta k_i = r_i$ from the global update $\Delta$ computed in Equation (4) requires proper alignment between key embeddings and residual embeddings. Methods lacking this alignment constraint struggle to separate the highly overlapping keys, leading to poor performance on this relation.

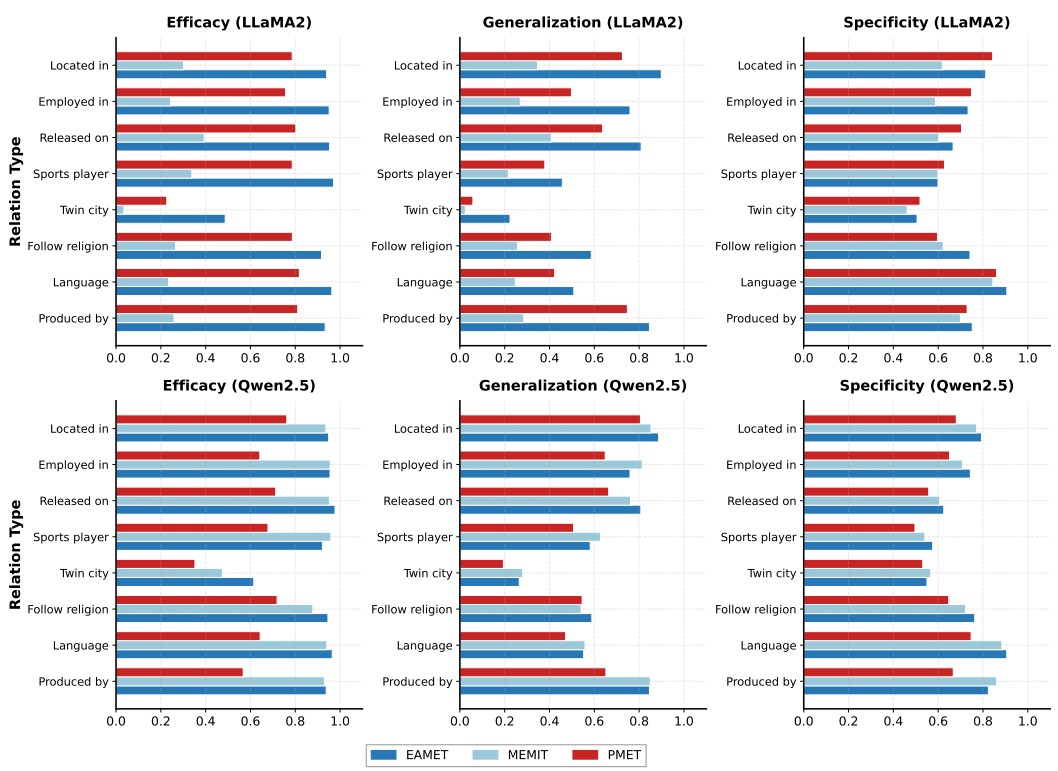

Figure 8: Performance comparison of different editing methods across different semantics.

## F.2 GENERAL ABILITY OF EDITED MODELS ON GLUE BENCHMARKS

***Additional Finding 2.* EAMET Better Preserves the General Ability of LLMs After Massive Editing.** We examine whether large-scale editing degrades the general capabilities of LLMs under MEMIT, PMET, and EAMET. Specifically, we evaluate three models (LLaMA2-7B, Qwen2.5-7B, and LLaMA3-8B) on the GLUE benchmark (Wang et al., 2018) after editing 10,000 knowledge facts from CounterFact and ZsRE. For reference, we also report the performance of the unedited models. As shown in Figure 9, EAMET consistently yields the smallest performance deviation from pre-edit baselines. On Qwen2.5-7B, the average deviation across six GLUE tasks is only 0.083 for CounterFact and 0.032 for ZsRE, substantially lower than MEMIT (0.266 and 0.349) and PMET (0.310 and 0.276). A similar trend holds for LLaMA3-8B and LLaMA2-7B: on CounterFact, EAMET achieves average deviations of 0.083 and 0.025, compared to 0.155 and 0.043 for MEMIT, the second-best method.

We attribute this robustness to EAMET's ability to extract more aligned memory representations across knowledge items. Such alignment reduces the likelihood of embedding inconsistency between key and residual spaces during massive editing, which may compromise the model's general capabilities. By mitigating this interference, EAMET effectively preserves the original functionality of the LLM.

## F.3 EVALUATION ON ADDITIONAL LLMS

We further evaluate the performance of EAMET on two additional LLMs: Gemma-7B and Phi-1.5. As shown in Table 8, EAMET consistently achieves the highest editing efficacy and generalization across both models and datasets. Moreover, it maintains competitive performance in terms of editing locality and generation ability.

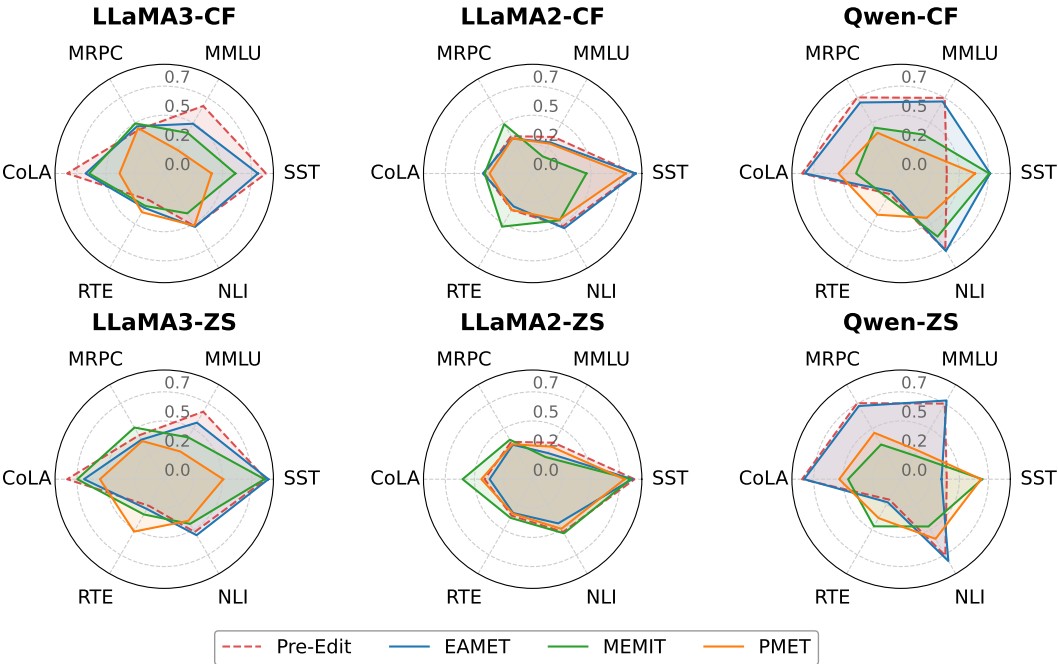

Figure 9: General ability of pre-edited model and models edited by different methods on GLUE benchmarks

Table 8: Performance comparison of different editing methods on Gemma-7B and Phi-1.5 on the Counterfact and ZsRE benchmarks.

| Model | Method | Counterfact | | | | ZsRE | | |
|---|---|---|---|---|---|---|---|---|
| | | Eff.↑ | Gen.↑ | Spe.↑ | Flu.↑ | Eff.↑ | Gen.↑ | Spe.↑ |
| | MEMIT | 93.01 | 54.33 | 74.88 | **538.92** | 84.61 | 73.66 | 23.04 |
| Gemma-7B | PMET | 91.69 | 47.24 | **75.22** | 533.62 | 83.05 | 73.95 | **23.85** |
| | **EAMET** | **95.29** | **68.53** | 70.22 | 530.93 | **91.69** | **86.43** | 23.37 |
| | MEMIT | 49.00 | 29.67 | 63.71 | 568.21 | 49.76 | 38.54 | 20.36 |
| Phi-1.5 | PMET | 38.49 | 20.75 | **67.01** | 579.19 | 32.00 | 24.47 | **21.54** |
| | **EAMET** | **67.76** | **40.13** | 62.08 | **580.92** | **60.61** | **45.42** | 21.29 |

## F.4 INTEGRATION WITH SEQUENTIAL EDITING

We further examine the impact of EAMET on sequential editing. We hypothesize that incorporating embedding alignment can increase the effective batch size at each step, thereby reducing the number of steps required to edit the same set of knowledge items. To this end, we adopt the state-of-the-art sequential editing method AlphaEdit, which preserves knowledge in LLMs by projecting updates onto the null space of preserved knowledge. To evaluate the benefit of embedding alignment, we replace AlphaEdit's target memory optimization with EAMET, resulting in a variant we call AlphaEdit-Aligned. Importantly, this substitution does not alter AlphaEdit's core design, since the method was not originally tailored for optimizing target memory. We then compare AlphaEdit and AlphaEdit-Aligned when editing 2,000 knowledge items on LLAMA2-7B from the Counterfact and ZsRE datasets, varying the batch size across 100, 200, 400, and 500 to evaluate how batch size influences performance.

***Additional Finding 3.*** **Integrating Embedding Alignment with Sequential Editing Enables Larger Batch Sizes.** As shown in Table 9, AlphaEdit-Aligned consistently outperforms AlphaEdit across all batch sizes, indicating that embedding alignment effectively enlarges the batch size per step. The improvement is especially pronounced on the Counterfact dataset, where batch editing

Table 9: Performance comparison of AlphaEdit and its version itegrated with EAMET on the Counterfact and ZsRE benchmarks.

| Method | Batch Size | Counterfact | | | | ZsRE | | |
|---|---|---|---|---|---|---|---|---|
| | | Eff.↑ | Gen.↑ | Spe.↑ | Flu.↑ | Eff.↑ | Gen.↑ | Spe.↑ |
| AlphaEdit | 100 | 49.10 | 39.13 | 61.01 | 331.83 | 95.75 | 87.75 | 17.05 |
| | 200 | 47.85 | 40.75 | 61.25 | 324.23 | 95.05 | 87.70 | 17.00 |
| | 400 | 41.55 | 37.03 | 59.51 | 228.53 | 94.80 | 86.75 | 16.80 |
| | 500 | 39.05 | 40.25 | 59.70 | 306.72 | 94.50 | 86.15 | 16.85 |
| AlphaEdit-Aligned | 100 | 96.75 | 66.13 | 66.48 | 505.59 | 96.55 | 87.75 | 17.00 |
| | 200 | 96.45 | 65.73 | 66.44 | 505.47 | 96.80 | 87.30 | 16.95 |
| | 400 | 96.45 | 64.85 | 66.33 | 505.82 | 95.61 | 86.95 | 16.80 |
| | 500 | 96.40 | 65.66 | 66.38 | 506.63 | 95.50 | 87.15 | 16.85 |

is notably more difficult without aligning key and residual embeddings. These results suggest that EAMET can be seamlessly integrated into sequential editing to further enhance editing performance.

## F.5 ABLATION STUDY

Table 10: Ablation study of EAMET components on Counterfact and ZsRE datasets.

| Method | Counterfact | | | | ZsRE | | |
|---|---|---|---|---|---|---|---|
| | Eff.↑ | Gen.↑ | Spe.↑ | Flu.↑ | Eff.↑ | Gen.↑ | Spe.↑ |
| EAMET (Full) | 89.09 | 61.21 | 72.19 | 519.42 | 89.47 | 81.34 | 15.70 |
| w/o KL Loss | 83.45 | 60.16 | 71.70 | 519.78 | 88.16 | 80.46 | 15.40 |
| w/o MSE Loss | 86.98 | 53.77 | 72.90 | 516.90 | 86.45 | 73.12 | 14.61 |

We further justify the design of combining KL loss and MSE loss by conducting ablations that remove either component. As shown in Table 10, the full version of EAMET consistently achieves the best overall performance across both datasets, while excluding either loss results in a clear performance drop. This confirms the effectiveness of our joint loss design.

Interestingly, the two losses exhibit different levels of importance depending on the dataset. On Counterfact, removing KL loss causes a 6% drop in editing efficacy, compared to only 2% when removing MSE loss. In contrast, on ZsRE, excluding KL loss leads to a minor 1% drop, whereas removing MSE loss results in a larger 3% decline. This difference stems from the structure of the datasets: in Counterfact, each knowledge item has a unique subject, making their key embeddings nearly orthogonal (low cosine similarity). Here, KL loss, which captures distributional differences across embeddings, plays a more critical role, while MSE contributes less. In ZsRE, however, many items share the same subject, leading to highly similar key embeddings (high cosine similarity). In this case, MSE loss is more important, as it directly aligns residual embeddings with their corresponding key embeddings within these subject-specific neighborhoods.

## F.6 EFFICIENCY ANALYSIS

We provide anaysis on the practical deployment cost of EAMET compared with MEMIT. We note that EAMET and MEMIT follow highly similar workflows for updating knowledge in LLMs: both require per-fact residual optimization. EAMET involves two additional steps: 1) the key embedding preparation stage, and 2) embedding alignment between the key and residual structures. We proceed to provide efficiency analysis on these two additional steps.

**The Cost of Key Embedding Preparation Stage.** We note that the key embedding preparation stage consists of two parts: 1) computing the key embeddings for all knowledge items to be edited at the target layer, and 2) computing the cosine similarities among all key embeddings. As shown in Table 11, retrieving key embeddings accounts for the majority of the time spent in the key prepa-

Table 11: Cost of key preparation steps as the number of edited facts increases.

| Number of Facts | Key Embeddings Cost | Similarities Cost |
|---|---|---|
| 10 | 1.1035 | 0.0015 |
| 100 | 10.6915 | 0.00153 |
| 1000 | 107.17 | 0.00034 |
| 2000 | 214.14 | 0.00025 |
| 5000 | 536.17 | 0.0007 |
| 10000 | 1076.18 | 0.00058 |

ration stage. Although computing key embeddings for 10,000 facts requires a nontrivial amount of time, this cost remains negligible (only about 1.8%) relative to the overall runtime of EAMET (59,154 s) and MEMIT (57,822 s) when editing 10,000 facts. In contrast, the runtime cost of computing pairwise cosine similarities among all key embeddings is trivial (below 0.002 s). This is because the operation can be efficiently executed by first normalizing all key embeddings to unit length and then performing dot-product computations, which are highly optimized on modern GPUs.

Table 12: Runtime and GPU memory cost for EAMET and MEMIT.

| Method | Optimizing 1 $r_i$ | | Editing 1 Fact | | Editing 100 Facts | |
|---|---|---|---|---|---|---|
| | Time | GPU | Time | GPU | Time | GPU |
| EAMET | 5.62s | 4.28GB | 26.98s | 7.41GB | 645.94s | 9.78GB |
| MEMIT | 5.59s | 4.07GB | 22.39s | 7.18GB | 636.94s | 9.18GB |

**The Cost of Embedding Alignment Stage.** As shown in Table 12, optimizing one residual in EAMET requires only an additional 0.03 seconds and 0.21 GB of memory compared to MEMIT. For the full editing of a single fact, EAMET incurs an extra 4.6 seconds, and this difference increases to 9 seconds when editing 100 facts. Although EAMET is slightly slower than MEMIT, the additional time and memory consumption are negligible, representing only 1.4% and 6.5% of MEMIT's overall cost, respectively. These results confirm that EAMET's improvements do not come at the expense of substantial deployment overhead; its runtime and resource requirements remain practical and comparable to MEMIT.

## F.7 RESULTS ON SMALL-SCALE EDITING

We further provide empirical results to demonstrate the performance of EAMET under single-edit or small-batch scenarios in LLaMA2-7B.

As shown in the table above, EAMET consistently outperforms MEMIT across all scales. Both methods perform similarly at 1 and 10 edits, achieving perfect efficacy with comparable generalization and specificity. However, once the number of edited facts exceeds 100, their performance diverges rapidly. Starting from 100 edits, EAMET maintains high quality (99.80% efficacy; 69.00% generalization), while MEMIT begins to decline (96.20%; 55.50%). As the scale grows to 1,000, 2,000, and 5,000 edits, EAMET continues to deliver strong results (94%-98% efficacy; 65%-68% generalization), whereas MEMIT degrades sharply, dropping from 49.98% and 36.25% at 1,000 edits to only 28.83% and 25.77% at 5,000 edits.

## F.8 ANALYSIS ON THE PERFORMANCE DIFFERENCE OF EAMET ACROSS LLMS AND DATASETS

Among the evaluated datasets (CounterFact, Wiki-recent, and ZsRE), Wiki-recent contains only 1,266 facts, whereas we edit 10,000 facts from each of the other two datasets. It is therefore expected that existing baseline methods also achieve relatively strong performance on Wiki-recent, although they still underperform EAMET.

Table 13: Performance comparison of EAMET and MEMIT across different numbers of edited facts on CounterFact and ZsRE datasets.

| Methods | # Facts | CounterFact | | | | ZsRE | | |
|---|---|---|---|---|---|---|---|---|
| | | Eff.↑ | Gen.↑ | Spe.↑ | Flu.↑ | Eff.↑ | Gen.↑ | Spe.↑ |
| EAMET | 5000 | 94.38 | 65.09 | 76.37 | 523.78 | 93.18 | 83.56 | 15.43 |
| | 2000 | 96.77 | 66.25 | 79.66 | 526.28 | 94.00 | 84.15 | 15.42 |
| | 1000 | 97.93 | 68.05 | 81.71 | 526.31 | 95.50 | 84.90 | 16.34 |
| | 100 | 99.80 | 69.00 | 82.83 | 525.23 | 96.00 | 87.00 | 15.40 |
| | 10 | 100.00 | 70.00 | 80.00 | 524.71 | 100.00 | 70.00 | 13.67 |
| | 1 | 100.00 | 100.00 | 100.00 | 524.59 | 100.00 | 100.00 | 0.00 |
| MEMIT | 5000 | 28.83 | 25.77 | 61.27 | 515.46 | 82.46 | 70.32 | 14.98 |
| | 2000 | 32.94 | 28.28 | 62.49 | 517.70 | 83.60 | 71.55 | 14.66 |
| | 1000 | 49.98 | 36.25 | 64.51 | 517.64 | 84.30 | 71.90 | 14.10 |
| | 100 | 96.20 | 55.50 | 84.55 | 519.38 | 85.00 | 74.00 | 14.67 |
| | 10 | 100.00 | 60.00 | 80.00 | 523.90 | 100.00 | 100.00 | 14.23 |
| | 1 | 100.00 | 100.00 | 100.00 | 524.30 | 100.00 | 60.00 | 0.00 |

For CounterFact and ZsRE, we observe that the performance gains of EAMET over prior methods differ across datasets. On CounterFact, the average improvement in editing efficacy over the second-best method is 8.01%, with the smallest improvement being 0.11%. On ZsRE, the average improvement increases to 14.48%, with the smallest improvement being 7.28%. We attribute this difference to how well each model generates distinct key embeddings for semantically unrelated facts. When key embeddings are not well separated and no alignment is enforced between key and residual embeddings, the reconstruction loss for individual facts inevitably increases.

To validate this argument, we analyze the cosine similarity among key embeddings generated by different models over 1,000 sampled facts from CounterFact and ZsRE. The table below reports the average cosine similarity for each model:

Table 14: Average cosine similarity among key embeddings for different models over 1,000 sampled facts from CounterFact and ZsRE.

| Models | CounterFact | ZsRE |
|---|---|---|
| LLaMA2-7B | 0.052843 | 0.048565 |
| Qwen-7B | 0.020466 | 0.022588 |
| DeepSeek-7B | 0.027811 | 0.029843 |
| Falcon-7B | 0.192273 | 0.196075 |

As shown in the table, different models exhibit varying inherent abilities to produce well-separated key embeddings. For LLaMA2-7B, key embeddings remain relatively entangled even for CounterFact, where each fact contains a distinct subject. This limited separation corresponds to lower MEMIT editing efficacy (24.95%), DeepSeek-7B exhibits a similar pattern, achieving 62.11% In contrast, Falcon-7B and Qwen-7B generate much more isolated key embeddings (0.1923 and 0.0205 on average), which aligns with their substantially higher MEMIT editing efficacy of 89.21% and 90.06%, respectively.

For ZsRE, many samples share identical subjects, making alignment between key and residual embeddings generally more challenging. Methods that do not enforce such alignment tend to struggle under this condition, leading to a more pronounced advantage for EAMET over prior approaches.

Overall, these results indicate that a model's inherent ability to generate well-separated key embeddings has considerable impact on editing performance. Despite these differences across models and datasets, EAMET consistently achieves the best results on all evaluated settings and LLMs.

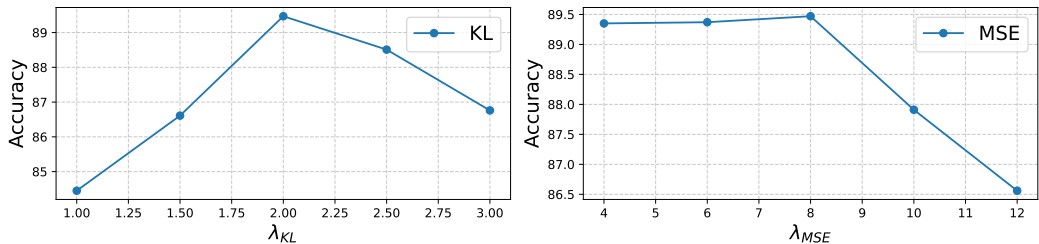

Figure 10: Impact of $\lambda_{KL}$ and $\lambda_{MSE}$ on EAMET's performance.

### F.9 DETAILED HYPERPARAMETER ANALYSIS

We analyze the impact of $\lambda_{KL}$ and $\lambda_{MSE}$ on EAMET's performance when editing 10,000 knowledge items from the ZsRE dataset. As shown in Figure 10, EAMET is more sensitive to the choice of $\lambda_{KL}$ than $\lambda_{MSE}$. A small $\lambda_{KL}$ weakens the alignment between residual and key embeddings, resulting in poor massive editing performance, whereas reducing $\lambda_{MSE}$ only causes a negligible drop in efficacy. The best performance is achieved when $\lambda_{KL} = 2$ and $\lambda_{MSE} = 8$, which are the hyperparameters adopted in the main paper. Increasing either weight beyond this point leads to decreased efficacy, as the optimization places less emphasis on updating new knowledge items.

Table 15: Impact of $M$ on EAMET's performance.

| $M$ | Counterfact | | | | ZsRE | | |
|---|---|---|---|---|---|---|---|
| | Eff.↑ | Gen.↑ | Spe.↑ | Flu.↑ | Eff.↑ | Gen.↑ | Spe.↑ |
| 5 | 87.23 | 54.74 | 73.95 | 517.58 | 88.10 | 79.56 | 15.51 |
| 10 | 87.96 | 57.58 | 74.25 | 517.38 | 88.89 | 79.62 | 15.63 |
| 50 | 89.09 | 61.21 | 73.69 | 519.89 | 89.47 | 81.34 | 15.70 |
| 100 | 86.17 | 86.85 | 74.52 | 517.01 | 89.02 | 81.14 | 15.70 |

We further analyze the impact of $M$, which is the number of cosine similarities selected for computing the MSE loss. As shown in Table 15, EAMET's editing performance on both datasets generally improves as M increases, reaching a peak around (M = 50), and then declines when M becomes too large. When M is small (e.g., M = 5), the alignment relies mainly on the KL-based distributional constraint, which enforces global structural consistency but does not guarantee precise value-level alignment between key embeddings and residual embeddings. Increasing M strengthens this value-based alignment and thus improves editing efficacy and generalization. However, when M becomes excessively large (e.g., M = 100), the dataset may not contain enough key embeddings that are meaningfully similar to the target key. As a result, the MSE loss becomes diluted across many low-relevance pairs, forcing the model to match less informative cosine similarities. This weakens the effectiveness of the alignment and causes a drop in overall performance.

***Additional Finding 4.*** **EAMET is Insensitive to The Choice of Hyperparameters.** EAMET introduces three additional hyperparameters: $\lambda_{KL}$, $\lambda_{MSE}$, and $M$. We analyze the impact of $\lambda_{KL}$ and $\lambda_{MSE}$ in Figure 10, and the influence of $M$ in Table 15. As shown in these results, when editing 10,000 CounterFact facts on LLaMA2-7B, different settings of $\lambda_{KL}$, $\lambda_{MSE}$, and $M$ do affect performance, but EAMET remains largely insensitive to the exact choice of values. For example, Figure 10 shows that setting $\lambda_{KL} = 2$ yields the lowest editing efficacy (85.3%). Although this is about 4% below the best-performing configuration, it still surpasses PMET and MEMIT by 11% and 51%, respectively. Likewise, varying $\lambda_{MSE}$ or $M$ changes performance by less than 3% between the best and worst cases. These findings demonstrate that EAMET is robust to the choice of hyperparameters.

## G SPECIFIC CASES

In this section, we present representative editing examples from the CounterFact dataset to qualitatively assess the generalization performance of different editing methods after editing 10,000 facts

on LLaMA2-7B and Qwen2.5-7B. These examples reveal that baseline methods often fail to generate the correct target answers or produce repetitive and incoherent responses. In contrast, EAMET consistently retrieves the intended factual content while producing fluent and informative completions, further validating the robustness and effectiveness of our proposed method.

| Model Editing Example on LLaMA2-7B | |
|---|---|
| Target Edit | The original language of The Icelandic Dream was |
| Original Object | `Icelandic` |
| Target Object | `Tamil` |
| **Generation Output** | |
| EAMET | The Icelandic Dream is full of people who speak `Tamil`, and the film is in `Tamil`. The film is about a family of `Tamil` refugees who are trying to make a new life for themselves in Europe. The film is about the struggles of the refugees, and the film is about the struggles of the people who are trying to make a |
| MEMIT | The Icelandic Dream is full of people who speak English, and the locals are very friendly. is is, a Belgian chain, has 100 restaurants in the UK, and is opening 10 more this year.,,, a Belgian chain, has 100 restaurants in the UK, and is opening 1 |
| PMET | The Icelandic Dream is full of people who speak in a language that is not English. The Icelandic Dream is a film about the immigrant experience. It is a film about the immigrant experience. It is a film about the immigrant experience. It is a film about the immigrant experience. It is a film about the |

| Model Editing Example on Qwen2.5-7B | |
|---|---|
| Editing Prompt | Argentine Football Association belongs to the organization of |
| Original Object | FIFA |
| Target Object | NATO |
| **Generation Output** | |
| EAMET | Argentine Football Association is part of NATO. NATO is military alliance. NATO is also called as what? NATO is a military alliance that was formed in 1949. It is also known as the North Atlantic Treaty Organization. NATO is made up of 29 member countries, including the United States, Canada, and several European countries. |
| MEMIT | Argentine Football Association is part of the American Football Association. It is a group of American players who are of African-American descent. The group was formed in 1991 and is based in the United States. The group is a member of the International Football Association. The group is also a member of the American Football Association. The group is also |
| PMET | Argentine Football Association is part of NATO's military alliance with the United States. The alliance is also known as the North Atlantic Treaty Organization or NATO. The alliance is a military alliance between the United States and NATO. The alliance is a military alliance between the United States and NATO. The alliance is a military alliance between the United States and NATO. The alliance [repetitive pattern] |

