# OpenReview forum: "EAMET: ROBUST MASSIVE MODEL EDITING VIA EMBEDDING ALIGNMENT OPTIMIZATION"
_ICLR.cc/2026/Conference — ICLR 2026 Poster_

### Official Review · Reviewer_jhJa · 2025-10-17

**Soundness:** 2
**Presentation:** 3
**Contribution:** 3
**Rating:** 6
**Confidence:** 5

**Summary:**

This paper aims to address the sharp performance degradation of large language models (LLMs) when performing massive simultaneous factual edits. The authors attribute this failure to embedding misalignment, a geometric inconsistency between the key embeddings representing knowledge and the residual embeddings responsible for executing updates, which leads to information loss during aggregated updates.

To tackle this issue, they propose EAMET, a novel model editing method whose core idea is to progressively and proactively align the structures of the key and residual embedding spaces during the optimization of each knowledge update, guided by KL divergence and MSE losses. Extensive experiments demonstrate that EAMET significantly outperforms existing methods, achieving higher accuracy and robustness when editing thousands of facts, particularly in challenging and realistic scenarios such as long-prefix interference and multi-point edits on the same subject.

**Strengths:**

* This paper attributes the performance degradation of existing model editing methods in large-scale realistic editing scenarios to **embedding misalignment**, which is a novel and interesting perspective.
* The paper conducts an in-depth theoretical and empirical analysis of **embedding misalignment**.
* The paper proposes **EAMET** to address the problem of **embedding misalignment**.
* Extensive experiments demonstrate the effectiveness of **EAMET**.

**Weaknesses:**

The assumption regarding **embedding misalignment** is overly strong, which concerns the theoretical foundation of the paper. Although **EAMET** shows strong experimental results, the theoretical aspect is an essential part of the paper’s contribution. Please refer to the **Questions** section for details.

**Questions:**

* In Equation (20), the paper assumes that in a large-scale editing batch, any knowledge update vector $(r_i)$ can be approximately represented as a weighted average of all other update vectors $(r_j)$ within the same batch. While this assumption might hold with a small $\epsilon_i$ in semantically related cases, it may not hold in semantically unrelated batches, where $\epsilon_i$ could be large. In such cases, the reconstruction residual would be excessively high and lose its interpretative significance. How do the authors explain this issue?
* Is the cosine similarity in Equation $9$ order-invariant? In massive editing, the order of knowledge updates should be arbitrary, so I believe an order-invariant definition should be provided here. If it is not order-invariant, what would the empirical results look like when the order is randomized?
* Why is preserving original knowledge defined as $\Delta C_p = 0$? Theoretically, $C_p$ should be a positive value to ensure the protection of existing knowledge. Practically speaking, removing $C_p$ should lead to a significant drop in editing performance (intuitively), since it serves as a regularization term, especially for preserving existing knowledge (specificity metric). How do the authors justify this design choice?
* Robustness when editing the same subject is indeed important. However, what happens if the facts about the same subject are potentially contradictory? Would the post-edit LLM produce inconsistent answers in such cases?
* Figure 1 is not mentioned in the main text, and the last case (i.e., {Sentences | $𝑠_i$ = Jeep Commander}) requires further clarification of its meaning to make it more reader-friendly.

---

> ### Author Response · Authors · 2025-11-17
> **Response by Authors to Reviewer jhJa (1/3)**
>
> We sincerely thank the reviewer for the time and effort dedicated to evaluating
> our paper. We address the concerns point by point below:
>
> **W1**: In Equation (20), the paper assumes that in a large-scale editing batch,
> any knowledge update vector $r_i $can be approximately represented as a weighted
> average of all other update vectors $r_j$ within the same batch. While this
> assumption might hold with a small $\epsilon_i$ in semantically related cases, it may not
> hold in semantically unrelated batches, where $\epsilon_i$ could be large. In such cases,
> the reconstruction residual would be excessively high and lose its
> interpretative significance. How do the authors explain this issue?
>
> **A1**: We thank the reviewer for the insightful question. We provide empirical
> results to demonstrate the performance of using a weighted average of all other
> update vectors $r_j$ to reconstruct the update vector $r_i$ on CounterFact and
> ZsRE, where samples within the CounterFact datasets shares distinct subjects and
> are thus semantically unrelated, whereas some samples within the ZsRE datasets
> shares the same subject and are thus more semantically related. Here, $r_i$ is
> computed using MEMIT on LLaMA2-7B. We report the average $\epsilon_i$ for each
> knowledge item $i$ in the table below:
>
> | Number of samples | CounterFact     | ZsRE     |
> |-------------------|--------|--------|
> | 10                | 0.9940 | 0.8553 |
> | 100               | 0.9651 | 0.8151 |
> | 1000              | 0.7929 | 0.6914 |
> | 2000              | 0.6257 | 0.5506 |
> | 5000              | 0.0122 | 0.0139 |
> | 10000             | 0.0011 | 0.0013 |
>
> As shown in the table above, the average value of $\epsilon_i$ decreases
> steadily as the number of editing samples increases. When the number of samples
> grows from 2,000 to 5,000, the average $\epsilon_i$ drops from 0.6257 to 0.0122
> on CounterFact, and from 0.5506 to 0.0139 on ZsRE. The value becomes nearly
> negligible when the number of samples reaches 10,000, which is the batch size
> used in our main experiments, achieving only 0.001 and 0.0013 for CounterFact and
> ZsRE, respectively.
>
> These results indicate that even for CounterFact, where facts are semantically
> unrelated, the update vector $r_i$ can be well approximated as a weighted
> average of other update vectors $r_j$ when the number of editing samples is
> sufficiently large. We argue that this phenomenon arises from the rich semantic
> information encoded in each $r_i$, which captures not only subject-specific
> information but also relational structure. With enough samples, this shared
> relational information enables accurate reconstruction of any individual update
> vector $r_i$ from the weighted combination of the remaining update vectors
> $r_j$.

---

> ### Author Response · Authors · 2025-11-17
> **Response by Authors to Reviewer jhJa (2/3)**
>
> **W2**: Is the cosine similarity in Equation 9 order-invariant? In massive
> editing, the order of knowledge updates should be arbitrary, so I believe an
> order-invariant definition should be provided here. If it is not
> order-invariant, what would the empirical results look like when the order is
> randomized?
>
> **A2**: We thank the reviewer for the insightful question. Yes, Equation 9 is
> order-invariant. We provide the detailed discussion in **Table 8** of **Appendix F.4**, which
> investigate the impact of different editing sequence on the performance of
> EAMET. We also recall the results here.
>
> | Method                       | **Counterfact** |        |        |        | **ZsRE** |        |        |
> |------------------------------|----------------------|--------|--------|--------|--------------|--------|--------|
> |                              | Eff. ↑              | Gen. ↑ | Spe. ↑ | Flu. ↑ | Eff. ↑       | Gen. ↑ | Spe. ↑ |
> | EAMET (original sequence)    | 89.09                | 61.21  | 72.19  | 519.06 | 89.47        | 81.34  | 15.70  |
> | -- random shuffle (seed=0)   | 88.21                | 60.79  | 71.84  | 519.21 | 87.63        | 77.42  | 15.56  |
> | -- random shuffle (seed=1)   | 89.11                | 60.78  | 72.03  | 518.84 | 86.99        | 76.08  | 15.58  |
> | -- random shuffle (seed=2)   | 88.91                | 59.38  | 72.34  | 518.23 | 87.56        | 77.47  | 15.59  |
>
> we examine EAMET's robustness under different editing orders on the Counterfact
> and ZsRE datasets. In Counterfact, all knowledge items have distinct subjects,
> whereas in ZsRE some items share the same subject and are adjacent in the
> original order. We therefore randomly shuffle the order of 10,000 items three
> times and report the average performance, alongside the original sequence as a
> reference.
>
> As shown in the above table, EAMET's performance remains stable across editing
> orders. On Counterfact, random shuffles produce only negligible variations in
> efficacy, generalization, and specificity. On ZsRE, editing efficacy shows a
> slight decline of about 2%, likely due to the neighborhood structure of items
> sharing the same subject in the original sequence. Overall, these results
> suggest that EAMET is largely insensitive to editing order, demonstrating strong
> robustness to sequence variations.
>
> **W3**: Why is preserving original knowledge defined as $\Delta C_p=0$?
> Theoretically, should be a positive value to ensure the protection of existing
> knowledge. Practically speaking, removing should lead to a significant drop in
> editing performance (intuitively), since it serves as a regularization term,
> especially for preserving existing knowledge (specificity metric). How do the
> authors justify this design choice?
>
> **A3**: We thank the reviewer for the insightful question. Let's recall that the
> update $\Delta$ is computed through:
>
> $$(W_0+\Delta)[K_p \quad K_t] = [M_p \quad M_t]$$
>
> where $W_0$ is the original parameters, $K_p$ is the keys of preserved
> knowledge, $K_t$ is the keys of target knowledge, $M_p$ is the memories of
> preserved knowledge, $M_t$ is the memories of target knowledge. We then multiply
> both sides by $[K_p \quad K_t]^T$ at the right hand side:
>
> $$(W_0+\Delta)[K_p \quad K_t] [K_p \quad K_t]^T = [M_p \quad M_t] [K_p \quad K_t]^T$$
> $$(W_0+\Delta)(K_pK_p^T + K_tK_t^T) = M_pK_p^T + M_tK_t^T$$
>
> As $W_0$ is the original parameters, we have $W_0K_p = M_p$. By subtracting
> $W_0K_pK_p^T=M_pK_p^T$ from both sides, we get:
>
> $$W_0K_tK_t^T+\Delta(K_pK_p^T + K_tK_t^T) = M_tK_t^T$$
>
> By organizing the above equation, we have:
>
> $$\Delta K_pK_p^T + (W_0+\Delta)K_tK_t^T = M_tK_t^T$$
>
> Recall that our goal is to update memory for target knowledge, that is:
>
> $$(W_0+\Delta)K_tK_t^T = M_tK_t^T$$
>
> Thus, we obtain
>
> $$\Delta K_p K_p^T = 0$$
>
> which indicates that the update $\Delta$ should induce no change to the
> preserved knowledge. Since the original model’s pretraining data is not
> accessible, we approximate $K_pK_p^T$ using a set of randomly sampled inputs
> from publicly available datasets.
>
> $$C_p = \lambda E_{k^p}[k_i^p (k_i^p)^T]$$
>
> Thus, *theoretically*, for a desired update $\Delta$, we would prefer $\Delta
> C_p = 0$. However, *empirically*, we do not enforce $\Delta C_p = 0$ and still
> use Equation 4 to compute the update $\Delta$. The assumption $\Delta C_p = 0$
> serves as an idealized condition that allows us to derive Theorem 1.

---

> ### Author Response · Authors · 2025-11-17
> **Response by Authors to Reviewer jhJa (3/3)**
>
> **W4**: Robustness when editing the same subject is indeed important. However,
> what happens if the facts about the same subject are potentially contradictory?
> Would the post-edit LLM produce inconsistent answers in such cases?
>
> **A4**: We thank the reviewer for the insightful question. To evaluate
> performance in the presence of conflicting information, we conduct an empirical
> study on LLaMA2-7B by editing 100 facts from the CounterFact dataset. To
> introduce conflicts, we replace the first five facts with duplicates of the same
> prompt, “The mother tongue of Danielle Darrieux is,” while assigning different
> target objects: “English,” “Spanish,” “Chinese,” “German,” and “Portuguese.”
>
> | Type          | Eff | Gen | Spe    | Flu     |
> |---------------|-----|-----|--------|---------|
> | Conflict      | 96  | 65  | 83.31  | 525.69  |
> | Non-Conflict  | 100 | 69  | 82.83  | 525.23  |
>
> As shown in the table above, model performance is only affected by the first
> five contradictory facts, where the final prediction becomes “English.” Despite
> this, EAMET successfully updates the remaining 95 facts while preserving the
> model’s general performance, as reflected by the stable fluency scores. This
> demonstrates that EAMET is robust even when the editing batch contains
> potentially conflicting information. In such cases, a model edited with EAMET
> still produces consistent answers for the contradictory facts while maintaining
> both the correctness of other edits and the overall model quality.
>
> In practice, however, the datasets used in our experiments have already been
> filtered to avoid such conflicts during editing. Ensuring the consistency and
> validity of the knowledge to be inserted remains an essential responsibility of
> anyone performing large-scale edits to LLMs.
>
> **W5**: Figure 1 is not mentioned in the main text, and the last case (i.e.,
> {Sentences | $s_i$ = Jeep Commander}) requires further clarification of its
> meaning to make it more reader-friendly.
>
> **A5**: We thank the reviewer for the valuable suggestion.
> {Sentences | $s_i$ = Jeep Commander} denotes the scenarios where multiple facts share the same subject.
> We will further clarify the meaning of the last case in Figure 1 in the revised manuscript.

---

> > ### Comment · Reviewer_jhJa · 2025-11-17
> >
> > Thank you for your detailed response. For A4, I would prefer to see results under different inputs that share the same semantics, rather than results under identical inputs.

---

> > > ### Author Response · Authors · 2025-11-18
> > >
> > > We apologize for the misunderstanding in A4. To further clarify, we provide
> > > additional empirical results using different input formulations that share the
> > > same underlying semantics. Following our previous setup, we introduce conflicts
> > > by replacing the second to fifth facts with paraphrased variants of the original
> > > fact, “The mother tongue of Danielle Darrieux is”. We construct two sets of
> > > paraphrased inputs:
> > >
> > > **Set 1: Minor paraphrases (same structure, different wording, same semantics)**
> > >
> > > This set preserves the syntactic structure of the original fact while varying
> > > the surface wording. The four paraphrases are:
> > >
> > > * “The native tongue of Danielle Darrieux is”
> > > * “The native language of Danielle Darrieux is”
> > > * “The mother language of Danielle Darrieux is”
> > > * “The first language of Danielle Darrieux is”
> > >
> > > **Set 2: Structural paraphrases (different structure, different wording, same semantics)**
> > >
> > > This set changes the syntactic structure more substantially while keeping the
> > > semantic meaning intact. The four paraphrases are:
> > >
> > > * “Danielle Darrieux speaks the language of”
> > > * “Danielle Darrieux, a native”
> > > * “What is the native language spoken by Danielle Darrieux?”
> > > * “Daniel Darrieux is a native speaker of”
> > >
> > > For both sets, we assign the target objects “Spanish,” “Chinese,” “German,” and
> > > “Portuguese” to the paraphrased facts, while keeping the original target object
> > > “English” unchanged for the first fact.
> > >
> > > We report in the tables above the performance of EAMET and MEMIT on LLaMA2-7B
> > > when editing 100 facts, where the first five facts are replaced with the
> > > conflicting paraphrased inputs described earlier. We also separately evaluate
> > > the editing efficacy of each individual conflicting fact.
> > >
> > > | Methods | Conflict Set | Eff. ↑ | Gen. ↑ | Spe. ↑ | Flu. ↑ |
> > > |---------|----------------|------|------|------|---------|
> > > | EAMET   | 1              | 96.00   | 64.50 | 83.20 | 524.84 |
> > > | | 2              | 96.00   | 63.50 | 83.40 | 525.16 |
> > > | MEMIT   | 1              | 93.70 | 58.00 | 83.40 | 522.06 |
> > > | | 2              | 93.00 | 52.00 | 84.10 | 523.83 |
> > >
> > >
> > > | Methods | Conflict Set | Fact ID | Eff  |
> > > |---------|--------------|---------|------|
> > > | EAMET   | 1            | 1       | 100    |
> > > |         |              | 2       | 0    |
> > > |         |              | 3       | 0    |
> > > |         |              | 4       | 0    |
> > > |         |              | 5       | 0    |
> > > | EAMET   | 2            | 1       | 100    |
> > > |         |              | 2       | 0    |
> > > |         |              | 3       | 0    |
> > > |         |              | 4       | 0    |
> > > |         |              | 5       | 0    |
> > > | MEMIT   | 1            | 1       | 100    |
> > > |         |              | 2       | 0    |
> > > |         |              | 3       | 0    |
> > > |         |              | 4       | 10    |
> > > |         |              | 5       | 0    |
> > > | MEMIT   | 2            | 1       | 100    |
> > > |         |              | 2       | 50    |
> > > |         |              | 3       | 0    |
> > > |         |              | 4       | 0    |
> > > |         |              | 5       | 0    |
> > >
> > > As shown in the results, EAMET’s performance remains stable across both conflict
> > > sets. The model edited by EAMET is only influenced by the first contradictory
> > > fact, whose final prediction consistently becomes “English”, which is the target
> > > of the first fact. Importantly, the edited model does not produce inconsistent
> > > predictions for the remaining paraphrased facts, even when their wording or
> > > syntactic structure differs greatly. For the other 95 facts, EAMET preserves
> > > strong editing efficacy and maintains the model’s general performance. In
> > > contrast, MEMIT is more sensitive to conflicts in the editing batch: its editing
> > > efficacy drops to 93.70% and 93.00% for the two conflict sets. Conflicting edits
> > > also lead to inconsistent outputs. For instance, for the second paraphrase (FACT
> > > ID 2) in Set 2, MEMIT produces the inconsistent object (“Spanish”) for 50% of
> > > the prefixed evaluation prompts.
> > >
> > > These findings highlight the robustness of EAMET
> > > when editing semantically similar yet conflicting facts. EAMET consistently
> > > produces stable predictions for inputs sharing the same underlying semantics,
> > > even when conflicting variants are included during optimization.

---

> > > > ### Comment · Reviewer_jhJa · 2025-11-20
> > > >
> > > > Thank you for your response. I chose to keep my score.

---

> > > > > ### Author Response · Authors · 2025-11-20
> > > > >
> > > > > Thank you for your response. We sincerely appreciate the time and effort you dedicated to reviewing our work. If there are any remaining aspects you would like to further discuss, we would be more than happy to engage.

---

### Official Review · Reviewer_QuKf · 2025-11-01

**Soundness:** 3
**Presentation:** 2
**Contribution:** 2
**Rating:** 4
**Confidence:** 4

**Summary:**

The paper studies why massive editing (e.g., 10k facts at once) breaks many model-editing methods and proposes EAMET, which aligns key and residual embedding spaces during the edit step. Under the stricter metric and in harder settings, EAMET claims high editing efficacy.

**Strengths:**

1. The problem is interesting where massive editing fails.
2. It introduces a stricter, more practical success metric tied to generation.
3. The results show the effectiveness of their method.

**Weaknesses:**

1. The primary experiments appropriately focus on large-batch editing, but the performance under single-edit or small-batch scenarios (e.g., editing only one or a few facts) remains unexplored. It would be valuable to examine whether the proposed alignment mechanism still provides benefits in these simpler settings.
2. The use of KL divergence over similarity-based softmax distributions to measure embedding misalignment is unconventional. The paper should clarify whether similar formulations have been used in prior literature.
3. The strategy for choosing the number of neighbors M in the pairwise MSE alignment term is not discussed.
4.The method introduces multiple additional hyperparameters, making the approach potentially fragile.
5. Although EAMET achieves notable gains in certain large-scale settings, its improvements are not consistently observed across all datasets and model families

**Questions:**

Please refer to the Weakness.

---

> ### Author Response · Authors · 2025-11-17
> **Response by Authors to Reviewer QuKf (1/3)**
>
> We sincerely thank the reviewer for the time and effort dedicated to evaluating
> our paper. We address the concerns point by point below:
>
> **W1**: The primary experiments appropriately focus on large-batch editing, but
> the performance under single-edit or small-batch scenarios (e.g., editing only
> one or a few facts) remains unexplored. It would be valuable to examine whether
> the proposed alignment mechanism still provides benefits in these simpler
> settings.
>
> **A1**: We thank the reviewer for the valuable suggestion on the exploration of
> the performance of EAMET under single-edit or small-batch scenarios. We further
> provide empirical results to demonstrate the performance of EAMET under
> single-edit or small-batch scenarios in LLaMA2-7B.
>
> | Methods | Number of Facts | CounterFact | | | | ZsRE | | | |
> |---------|-----------------|-----------------|-----|-----|------|-----------|------|------|------|
> |         |                 | **Eff** | **Gen** | **Spe** | **Flu** | **Eff** | **Gen** | **Spe** | **Flu** |
> | **EAMET** | 5000 | 94.38 | 65.09 | 76.37 | 523.78 | 93.18 | 83.56 | 15.43 | 523.78 |
> |          | 2000 | 96.77 | 66.25 | 79.66 | 526.28 | 94.00 | 84.15 | 15.42 | 526.28 |
> |          | 1000 | 97.93 | 68.05 | 81.71 | 526.31 | 95.50 | 84.90 | 16.34 | 526.31 |
> |          | 100  | 99.80 | 69.00 | 82.83 | 525.23 | 96.00 | 87.00 | 15.40 | 525.23 |
> |          | 10   | 100.00 | 70.00 | 80.00 | 524.71 | 100.00 | 70.00 | 13.67 | 524.71 |
> |          | 1    | 100.00 | 100.00 | 100.00 | 524.59 | 100.00 | 100.00 | 0.00 | 524.59 |
> | **MEMIT** | 5000 | 28.83 | 25.77 | 61.27 | 515.46 | 82.46 | 70.32 | 14.98 | 515.46 |
> |           | 2000 | 32.94 | 28.28 | 62.49 | 517.70 | 83.60 | 71.55 | 14.66 | 517.70 |
> |           | 1000 | 49.98 | 36.25 | 64.51 | 517.64 | 84.30 | 71.90 | 14.10 | 517.64 |
> |           | 100  | 96.20 | 55.50 | 84.55 | 519.38 | 85.00 | 74.00 | 14.67 | 519.38 |
> |           | 10   | 100.00 | 60.00 | 80.00 | 523.90 | 100.00 | 100.00 | 14.23 | 523.90 |
> |           | 1    | 100.00 | 100.00 | 100.00 | 524.30 | 100.00 | 60.00 | 0.00 | 524.30 |
>
> As shown in the table above, EAMET consistently outperforms MEMIT across all
> scales. Both methods perform similarly at 1–10 edits, achieving perfect efficacy
> with comparable generalization and specificity. However, once the number of
> edited facts exceeds 100, their performance diverges rapidly. Starting from 100
> edits, EAMET maintains high quality (99.80% efficacy; 69.00% generalization),
> while MEMIT begins to decline (96.20%; 55.50%). As the scale grows to 1,000,
> 2,000, and 5,000 edits, EAMET continues to deliver strong results (94–98%
> efficacy; 65–68% generalization), whereas MEMIT degrades sharply, dropping from
> 49.98% / 36.25% at 1,000 edits to only 28.83% / 25.77% at 5,000 edits.
>
> **W2**: The use of KL divergence over similarity-based softmax distributions to
> measure embedding misalignment is unconventional. The paper should clarify
> whether similar formulations have been used in prior literature.
>
> **A2**: We appreciate the reviewer’s insightful question. Although directly
> applying KL divergence to similarity-based distributions is not the dominant
> choice in knowledge editing literature, the formulation is conceptually aligned
> with established representation-learning methods. Prior works in contrastive
> learning [1,2] (e.g., InfoNCE loss [1]) and distributional alignment [3]
> routinely convert similarity scores into probability distributions using softmax
> and then optimize divergence-based objectives. Our method adopts the same
> principle: normalizing similarities creates a distribution that reflects the
> relative geometry of all key embeddings, and KL divergence provides a sensitive
> measure for enforcing this structural consistency.
>
> [1] Chen, Ting, et al. "A simple framework for contrastive learning of visual
> representations." International conference on machine learning. PmLR, 2020.
>
> [2] He, Kaiming, et al. "Momentum contrast for unsupervised visual
> representation learning." Proceedings of the IEEE/CVF conference on computer
> vision and pattern recognition. 2020.
>
> [3] Sun, Baochen, and Kate Saenko. "Deep coral: Correlation alignment for deep
> domain adaptation." European conference on computer vision. Cham: Springer
> International Publishing, 2016.

---

> ### Author Response · Authors · 2025-11-17
> **Response by Authors to Reviewer QuKf (2/3)**
>
> **W3**: The strategy for choosing the number of neighbors M in the pairwise MSE
> alignment term is not discussed.
>
> **A3**: We thank the reviewer for the valuable suggestion on the strategy for
> choosing the number of neighbors M in the pairwise MSE alignment term. We
> further provide the empirical results to demonstrate the performance of EAMET
> with different values of M when editing 10000 facts on LLaMA2-7B.
>
> | M | CounterFact | | | | ZsRE | | |
> |---|--------|------|------|-------|--------|------|------|
> |   | **Eff** | **Gen** | **Spe** | **Flu** | **Eff** | **Gen** | **Spe** |
> | 5   | 87.23 | 54.74 | 73.95 | 517.58 | 88.10 | 79.56 | 15.51 |
> | 10  | 87.96 | 57.58 | 74.25 | 517.38 | 88.89 | 79.62 | 15.63 |
> | 50  | 89.09 | 61.21 | 73.69 | 519.89 | 89.47 | 81.34 | 15.70 |
> | 100 | 86.17 | 54.72 | 74.52 | 517.01 | 89.02 | 81.14 | 15.70 |
>
> As shown in the table above, EAMET’s editing performance on both datasets
> generally improves as M increases, reaching a peak around (M = 50), and then
> declines when M becomes too large. When M is small (e.g., M = 5), the alignment
> relies mainly on the KL-based distributional constraint, which enforces global
> structural consistency but does not guarantee precise value-level alignment
> between key embeddings and residual embeddings. Increasing M strengthens this
> value-based alignment and thus improves editing efficacy and generalization.
> However, when M becomes excessively large (e.g., M = 100), the dataset may not
> contain enough key embeddings that are meaningfully similar to the target key.
> As a result, the MSE loss becomes diluted across many low-relevance pairs,
> forcing the model to match less informative cosine similarities. This weakens
> the effectiveness of the alignment and causes a drop in overall performance.
>
> **W4**: The method introduces multiple additional hyperparameters, making the
> approach potentially fragile.
>
> **A4**: We thank the reviewer for the insightful question. EAMET introduces
> three additional hyperparameters: $\lambda_{\text{KL}}$, $\lambda_{\text{MSE}}$,
> and $M$. We analyze the impact of $\lambda_{\text{KL}}$ and
> $\lambda_{\text{MSE}}$ in **Figure 10** of **Appendix F.7**, and the influence of $M$ in
> the table above. As shown in these results, when editing 10,000 CounterFact
> facts on LLaMA2-7B, different settings of $\lambda_{\text{KL}}$,
> $\lambda_{\text{MSE}}$, and $M$ do affect performance, but EAMET remains largely
> insensitive to the exact choice of values. For example, Figure 10 shows that
> setting $\lambda_{\text{KL}} = 2$ yields the lowest editing efficacy (85.3%).
> Although this is about 4% below the best-performing configuration, it still
> surpasses PMET and MEMIT by 11% and 51%, respectively. Likewise, varying
> $\lambda_{\text{MSE}}$ or $M$ changes performance by less than 3% between the
> best and worst cases. These findings demonstrate that EAMET is robust to the
> choice of hyperparameters. We will integrate this discussion more clearly into
> the revised manuscript.

---

> ### Author Response · Authors · 2025-11-17
> **Response by Authors to Reviewer QuKf (3/3)**
>
> **W5**: Although EAMET achieves notable gains in certain large-scale settings,
> its improvements are not consistently observed across all datasets and model
> families.
>
> **A5**: We thank the reviewer for the insightful observation. Among the
> evaluated datasets (CounterFact, Wiki-recent, and ZsRE), Wiki-recent contains
> only 1,266 facts, whereas we edit 10,000 facts from each of the other two
> datasets. It is therefore expected that existing baseline methods also achieve
> relatively strong performance on Wiki-recent, although they still underperform
> EAMET.
>
> For CounterFact and ZsRE, we observe that the performance gains of EAMET over
> prior methods differ across datasets. On CounterFact, the average improvement in
> editing efficacy over the second-best method is 8.01%, with the smallest
> improvement being 0.11%. On ZsRE, the average improvement increases to 14.48%,
> with the smallest improvement being 7.28%. We attribute this difference to how
> well each model generates distinct key embeddings for semantically unrelated
> facts. When key embeddings are not well separated and no alignment is enforced
> between key and residual embeddings, the reconstruction loss for individual
> facts inevitably increases.
>
> To validate this argument, we analyze the cosine similarity among key embeddings
> generated by different models over 1,000 sampled facts from CounterFact and
> ZsRE. The table below reports the average cosine similarity for each model:
>
> | Models      | CounterFact | ZsRE     |
> | ----------- | ----------- | -------- |
> | LLaMA2-7B   | 0.052843    | 0.048565 |
> | Qwen-7B     | 0.020466    | 0.022588 |
> | DeepSeek-7B | 0.027811    | 0.029843 |
> | Falcon-7B   | 0.192273    | 0.196075 |
>
> As shown in the table, different models exhibit varying inherent abilities to
> produce well-separated key embeddings. For LLaMA2-7B, key embeddings remain
> relatively entangled even for CounterFact, where each fact contains a distinct
> subject. This limited separation corresponds to lower MEMIT editing efficacy
> (24.95%). DeepSeek-7B exhibits a similar pattern, achieving 62.11%. In contrast,
> Falcon-7B and Qwen-7B generate much more isolated key embeddings (0.1923 and
> 0.0205 on average), which aligns with their substantially higher MEMIT editing
> efficacy of 89.21% and 90.06%, respectively.
>
> For ZsRE, many samples share identical subjects, making alignment between key
> and residual embeddings generally more challenging. Methods that do not enforce
> such alignment tend to struggle under this condition, leading to a more
> pronounced advantage for EAMET over prior approaches.
>
> Overall, these results indicate that a model’s inherent ability to generate
> well-separated key embeddings has considerable impact on editing performance.
> Despite these differences across models and datasets, EAMET consistently
> achieves the best results on all evaluated settings and LLMs.

---

> ### Author Response · Authors · 2025-11-26
>
> Considering the upcoming deadlines, we would be immensely grateful if you could
> provide any additional feedback or clarifications at your earliest convenience.
> Your guidance is crucial for the improvement of our work, and we deeply
> appreciate the time and effort you have invested in reviewing our responses.
>
> We sincerely value your invaluable feedback and have made substantial revisions
> and provided detailed responses to address your concerns in both the rebuttal
> and the revised version. We have already include **W1&A1** into **Appendix F.6**,
> **W3&A3** and **W4&A4** into **Appendix F.9**, and **W5&A5** into **Appendix F.8** in
> the revised version. We have also added corresponding citations for the use of
> KL divergence over similarity-based softmax distributions to measure embedding
> misalignment for **W2&A2** at **Line 308**.

---

### Official Review · Reviewer_XmZ3 · 2025-11-03

**Soundness:** 3
**Presentation:** 2
**Contribution:** 3
**Rating:** 6
**Confidence:** 4

**Summary:**

The paper introduces an innovative approach, EAMET, for enhancing the robustness of large-scale model editing, focusing on aligning key and residual embeddings to improve the editing efficacy in large language models. The research addresses a pertinent issue in model editing, and the proposed method demonstrates promising results across various benchmarks. However, there are several areas that need further clarification and improvement for better understanding and impact.

**Strengths:**

1. The proposed EAMET framework brings an innovative solution to improve the effectiveness of model editing, particularly in massive editing scenarios. By aligning key and residual embeddings, it overcomes limitations seen in traditional methods, making it a valuable contribution to the field.

2. The paper includes comprehensive experiments on multiple datasets and models, demonstrating the effectiveness of the proposed method in real-world scenarios. The experimental design is solid, and the results are promising, showing EAMET’s superiority over existing methods.

**Weaknesses:**

1. The current empirical analysis (starting at line 200)  would benefit from a deeper investigation into how the success rate of editing varies across different categories of knowledge, particularly considering their varying degrees of representation inconsistency. This would provide stronger evidence for the challenges addressed by the proposed method.

2. The paper does not provide a detailed complexity analysis of the Key Embedding Preparation step, which involves calculating a large number of cosine similarities.

3. While not a major issue, some points in the writing need improvement. For example, the abbreviation for "CF ZS" in the experiments section is not defined; Formula (15) lacks punctuation; The definition of "N" in Formula (14) is not provided.

4. Formula (14) is central to the paper’s approach, but it is difficult to fully understand without a more detailed explanation. While I can infer its meaning, the explanation provided is insufficient, and I spent considerable time trying to understand it.

**Questions:**

See weakness.

---

> ### Author Response · Authors · 2025-11-17
> **Response by Authors to Reviewer XmZ3 (1/2)**
>
> We sincerely thank the reviewer for the time and effort dedicated to evaluating
> our paper. We address the concerns point by point below:
>
> **W1**: The current empirical analysis (starting at line 200) would benefit from
> a deeper investigation into how the success rate of editing varies across
> different categories of knowledge, particularly considering their varying
> degrees of representation inconsistency. This would provide stronger evidence
> for the challenges addressed by the proposed method.
>
> **A1**: We thank the reviewer for the valuable suggestion on the deeper
> discussion of how the success rate of editing varies across different categories
> of knowledge. We provide the detailed discussion in **Figure 8** of **Appendix
> F.1** which demonstrates the editing performance of EAMET in comparison with
> existing methods across different categories of knowledge.
>
> We extract samples with specific relation types from the CounterFact dataset to
> evaluate the performance of different editing methods across semantic
> categories. As shown in **Figure 8**, EAMET consistently achieves the highest
> editing efficacy and generalization on both evaluated models for most semantic
> types. On LLaMA2-7B, EAMET outperforms the second-best method (PMET) by
> approximately 10% in efficacy and 20% in generalization.
>
> We observe an interesting phenomenon on the **twin-city** relation of LLaMA2-7B:
> EAMET achieves **2× higher efficacy** and **4× higher generalization** compared
> to PMET, while MEMIT nearly fails on this relation, yielding efficacy and
> generalization scores close to 0%. This occurs because facts involving the
> twin-city relation are typically expressed in forms such as “The twin city of
> {subject}” or “What is the twin city of {subject}?”. The key embeddings, which
> are extracted from the last subject token, are therefore highly similar across
> facts due to the shared prefixes in these templates. As a result, reconstructing
> each individual update $\Delta k_i = r_i$ from the global update $\Delta$
> computed in **Equation (7)** requires proper alignment between key embeddings
> and residual embeddings. Methods lacking this alignment constraint struggle to
> separate the highly overlapping keys, leading to poor performance on this
> relation.
>
> We will incorporate this additional discussion into the revised manuscript.
>
>
> **W2**: The paper does not provide a detailed complexity analysis of the Key
> Embedding Preparation step, which involves calculating a large number of cosine
> similarities.
>
> **A2**: We thank the reviewer for the valuable suggestion on the complexity
> analysis of the Key Embedding Preparation step. We note that the key embedding
> preparation stage consists of two parts: 1) computing the key embeddings for all
> knowledge items to be edited at the target layer, and 2) computing the cosine
> similarities among all key embeddings. We further provide the detailed
> runtime cost of these two parts in the following table:
>
> | Number of Edited facts | Compute Key Embeddings Cost | Similarities Computation Cost |
> |----------|------------------|-------------|
> | 10       | 1.1035           | 0.0015      |
> | 100      | 10.6915          | 0.00153     |
> | 1000     | 107.17           | 0.00034     |
> | 2000     | 214.14           | 0.00025     |
> | 5000     | 536.17           | 0.0007      |
> | 10000    | 1076.18          | 0.00058     |
>
> As shown in the table above, retrieving key embeddings accounts for the majority
> of the time spent in the key preparation stage. Although computing key
> embeddings for 10,000 facts requires a nontrivial amount of time, this cost
> remains negligible (only about 1.8%) relative to the overall runtime of EAMET
> (59,154 s) and MEMIT (57,822 s) when editing 10,000 facts. In contrast, the
> runtime cost of computing pairwise cosine similarities among all key embeddings
> is trivial (below 0.002 s). This is because the operation can be efficiently
> executed by first normalizing all key embeddings to unit length and then
> performing dot-product computations, which are highly optimized on modern GPUs.
> We will further incorporate the discussion into the revised manuscript.

---

> ### Author Response · Authors · 2025-11-17
> **Response by Authors to Reviewer XmZ3 (2/2)**
>
> **W3**: While not a major issue, some points in the writing need improvement.
> For example, the abbreviation for "CF ZS" in the experiments section is not
> defined; Formula (15) lacks punctuation; The definition of "N" in Formula (14)
> is not provided.
>
> **A3**: We thank the reviewer for the valuable suggestion on the writing improvement. Here, "CF" and "ZS" stands for "Counterfact" and "ZsRE" datasets. Formula (15) should be revised as
>
> $Eff=E_i[o_i=\arg\mathop{\max}_{o} P_{f_\theta}(o\ |\ (s_i,rel_i))]$.
>
> And the definition of "$N_{FP}$" in Formula (14) is the number of random prefixes $f_i$ used to construct inputs. We will further clarify these points in the revised manuscript.
>
> **W4**: Formula (14) is central to the paper’s approach, but it is difficult to
> fully understand without a more detailed explanation. While I can infer its
> meaning, the explanation provided is insufficient, and I spent considerable time
> trying to understand it.
>
> **A4**: We thank the reviewer for the valuable suggestion regarding the
> clarification of Formula (14). Formula (14) defines the optimization objective
> for computing the target residual embedding $r_i$, which is added to the hidden
> state $h_i^L$ when editing the factual association $(s_i, rel_i, o_i)$.
> Intuitively, the objective consists of two components working together:
>
> 1. **Task Loss:** The negative log-likelihood term $\log P(o_i \mid s_i, rel_i,
>    r_i)$ ensures that adding the residual $r_i$ effectively updates the model
>    so that it outputs the correct target object $o_i$.
>
> 2. **Alignment Regularization:** The additional regularization terms enforce
>    that the embedding space of the optimized residuals $\{r_i\}$ remains aligned
>    with that of the key embeddings $\{k_i\}$. This alignment preserves semantic
>    consistency, ensuring that “similar keys receive similar residual updates,”
>    which prevents the reconstruction failure for individual facts that may occur
>    during large-scale editing.
>
> We will incorporate this intuitive explanation alongside the mathematical formulation of Formula (14) in the revised manuscript.

---

> ### Author Response · Authors · 2025-11-26
>
> Considering the upcoming deadlines, we would be immensely grateful if you could
> provide any additional feedback or clarifications at your earliest convenience.
> Your guidance is crucial for the improvement of our work, and we deeply
> appreciate the time and effort you have invested in reviewing our responses.
>
> We sincerely value your invaluable feedback and have made substantial revisions
> and provided detailed responses to address your concerns in both the rebuttal
> and the revised version. We have already include **W1&A1** into **Appendix F.1**,
> **W2&A2** into **Appendix F.6** in the revised version. We also revised **Table 3**,
> **Formula (14)** and **(15)** accordingly.

---

### Official Review · Reviewer_ifDP · 2025-11-10

**Soundness:** 3
**Presentation:** 3
**Contribution:** 2
**Rating:** 6
**Confidence:** 2

**Summary:**

This paper proposes EAMET, a massive model editing method that aims to address a key failure of existing locate-then-edit approaches such as MEMIT/PMET: the embedding misalignment between key embeddings and residual/memory embeddings when many facts are edited simultaneously. The authors provide a theoretical analysis linking reconstruction error in closed-form edits to the mismatch between similarity structures in key space and residual space, then introduce an alignment-based optimization of residual embeddings integrated into a MEMIT-style update. Empirically, EAMET seems to show consistently improved efficacy, robustness, and portability across multiple LLM architectures and factual datasets, with minimal degradation to general capabilities.

**Strengths:**

+ This paper clearly identifies and theoretically characterizes embedding misalignment as a core scalability bottleneck for existing massive edits approaches.

+ The proposed EAMET is architecturally compatible with MEMIT-style pipelines. It introduces alignment-based optimization of the derived residual embeddings, which makes sense to address the embedding misalignment problem.

+ The experiments with 10k+ edits or long prefixes seem to demonstrate the effectiveness of the proposed method.

**Weaknesses:**

- It seems that the per-fact residual optimization and alignment steps may be expensive at very large scales. I'm curious about the detailed runtime, memory, and scalability trade-offs with the MEMIT-style baselines.

- The optimization of the alignment seems to be sequential. I also wonder if the optimization order can have a difference to the editing results.

**Questions:**

Please refer to my summary of weaknesses.

---

> ### Author Response · Authors · 2025-11-17
> **Response by Authors to Reviewer ifDP**
>
> We sincerely thank the reviewer for the time and effort dedicated to evaluating
> our paper. We address the concerns point by point below:
>
> **W1**: It seems that the per-fact residual optimization and alignment steps may be
> expensive at very large scales. I'm curious about the detailed runtime, memory,
> and scalability trade-offs with the MEMIT-style baselines.
>
> **A1**: We thank the reviewer for the valuable suggestion on the practical
> deployment cost of EAMET compared with MEMIT. We note that EAMET and MEMIT
> follow highly similar workflows for updating knowledge in LLMs: both require
> per-fact residual optimization, and the only additional step in EAMET is the
> embedding alignment between the key and residual structures. To further clarify
> this point, we provide empirical measurements showing that EAMET incurs nearly
> identical runtime and memory overhead compared to MEMIT.
>
> | Method | Optimizing 1 Residual |     | Editing 1 Fact           |     | Editing 100 Facts            |     |
> |--------|-----------------------|-----|--------------------------|-----|------------------------------|-----|
> |        | time                  | GPU | time | GPU | time   | GPU |
> | EAMET  | 5.62s              | 4.28GB| 26.98s                  | 7.41 GB| 645.94s                       | 9.78GB|
> | MEMIT  | 5.59s              | 4.07GB| 22.39s                  | 7.18 GB| 636.94s                       | 9.18GB|
>
> As shown in the above Table, optimizing one residual in EAMET requires only an
> additional 0.03 seconds and 0.21 GB of memory compared to MEMIT. For the full
> editing of a single fact, EAMET incurs an extra 4.6 seconds, and this difference
> increases to 9 seconds when editing 100 facts. Although EAMET is slightly slower
> than MEMIT, the additional time and memory consumption are negligible,
> representing only 1.4% and 6.5% of MEMIT’s overall cost, respectively. These
> results confirm that EAMET’s improvements do not come at the expense of
> substantial deployment overhead; its runtime and resource requirements remain
> practical and comparable to MEMIT.
>
> **W2**: The optimization of the alignment seems to be sequential. I also wonder if
> the optimization order can have a difference to the editing results
>
> **A2**: We thank the reviewer for pointing out the concerns on the robustness of
> EAMET to different editing sequence. We already provide discussion in **Table 8**
> of **Appendix F.4**, which investigate the impact of different editing sequence
> on the performance of EAMET. We also recall the results here.
>
> | Method                       | **Counterfact** |        |        |        | **ZsRE** |        |        |
> |------------------------------|----------------------|--------|--------|--------|--------------|--------|--------|
> |                              | Eff. ↑              | Gen. ↑ | Spe. ↑ | Flu. ↑ | Eff. ↑       | Gen. ↑ | Spe. ↑ |
> | EAMET (original sequence)    | 89.09                | 61.21  | 72.19  | 519.06 | 89.47        | 81.34  | 15.70  |
> | -- random shuffle (seed=0)   | 88.21                | 60.79  | 71.84  | 519.21 | 87.63        | 77.42  | 15.56  |
> | -- random shuffle (seed=1)   | 89.11                | 60.78  | 72.03  | 518.84 | 86.99        | 76.08  | 15.58  |
> | -- random shuffle (seed=2)   | 88.91                | 59.38  | 72.34  | 518.23 | 87.56        | 77.47  | 15.59  |
>
> we examine EAMET's robustness under different editing orders on the Counterfact
> and ZsRE datasets. In Counterfact, all knowledge items have distinct subjects,
> whereas in ZsRE some items share the same subject and are adjacent in the
> original order. We therefore randomly shuffle the order of 10,000 items three
> times and report the average performance, alongside the original sequence as a
> reference.
>
> As shown in the above table, EAMET's performance remains stable across editing
> orders. On Counterfact, random shuffles produce only negligible variations in
> efficacy, generalization, and specificity. On ZsRE, editing efficacy shows a
> slight decline of about 2%, likely due to the neighborhood structure of items
> sharing the same subject in the original sequence. Overall, these results
> suggest that EAMET is largely insensitive to editing order, demonstrating strong
> robustness to sequence variations.

---

> ### Author Response · Authors · 2025-11-26
>
> Considering the upcoming deadlines, we would be immensely grateful if you could
> provide any additional feedback or clarifications at your earliest convenience.
> Your guidance is crucial for the improvement of our work, and we deeply
> appreciate the time and effort you have invested in reviewing our responses.
>
> We sincerely value your invaluable feedback and have made substantial revisions
> and provided detailed responses to address your concerns in both the rebuttal
> and the revised version. We have already include **W1&A1** into **Appendix F.6**,
> and we moved **W2&A2** to **Section 6.3** in the revised version.

---

### Author Response · Authors · 2025-11-17
**Global Response by Authors**

Dear Reviewers,

We thank all reviewers for their insightful feedback. The comments have been invaluable in improving our work. In this rebuttal, we provide new experimental results (including new correlation, reliability, and variance analyses) to address all concerns. All these new analyses and the corresponding discussions will be integrated into our revised paper.

Best regards

---

### Author Response · Authors · 2025-11-26
**Global Response by Authors**

Dear Reviewers,

We again thank all reviewers for their insightful feedback. In response to your
comments, we have diligently revised our paper and have uploaded the new version
as a revised PDF. For ease of reference, we have highlighted the changes in red.

Considering the upcoming deadlines, we would be immensely grateful if you could
provide any additional feedback or clarifications at your earliest convenience.
Your guidance is crucial for the improvement of our work, and we deeply
appreciate the time and effort you have invested in reviewing our responses.

Best regards

---

### Author Response · Authors · 2025-11-30
**Summary of Our Responses (2/2)**

**IC5 – Performance differences across datasets and model families (XmZ3–W5)**

We added a new analysis that explains performance differences across datasets
and models through the lens of cosine similarity among key embeddings. Our
results show that a model’s inherent ability to produce well-separated key
embeddings strongly impacts editing performance: models with more entangled keys
make reconstruction more difficult for methods without alignment, while EAMET
remains robust.

Despite these variations, EAMET consistently achieves the best performance
across all evaluated datasets and LLMs. We incorporate this discussion in
**Appendix F.8** of the revised manuscript.


**IC6 – Explanation of Equation (20) in semantically unrelated batches
(jhJa–W1)**

We provide empirical evidence showing that, even for semantically unrelated
batches (e.g., CounterFact with distinct subjects), the reconstruction error
($\epsilon_i$) becomes very small when the number of editing samples is
sufficiently large. This validates the approximation that an update vector ($r_i$)
can be represented as a weighted average of other update vectors when the batch
size is large.


**IC7 – Implication of the assumption ($\Delta C_p = 0$) (jhJa–W3)**

We added a detailed derivation explaining why ($\Delta C_p = 0$) arises as a
desirable property for an ideal update ($\Delta$). Theoretically, the condition
($\Delta C_p = 0$) characterizes the case where preserved knowledge remains
unchanged. In practice, however, we do **not** explicitly enforce ($\Delta C_p =
0$); instead, we still compute ($\Delta$) using Eq. (4), and the assumption is
used as an idealized condition for deriving Theorem 1.


**IC8 – Behavior of EAMET under conflicting information for the same subject
(jhJa–W4)**

We conducted additional experiments to study EAMET’s behavior when the editing
batch contains potentially conflicting information for the same subject. We
consider two scenarios:

1. **Different targets for identical inputs**, and
2. **Different targets for paraphrased inputs with the same underlying semantics**.

For the first scenario, we show that EAMET converges to a consistent prediction
for the conflicting facts while successfully updating the remaining facts and
preserving overall model quality. For the second scenario, using both minor and
structural paraphrases, the edited model does not produce inconsistent
predictions across paraphrases, even when their surface forms differ
substantially.

These experiments demonstrate that EAMET is robust to conflicting updates and
maintains semantic consistency across paraphrased inputs.

Once again, we sincerely thank the Area Chair and all reviewers for their
careful evaluation and valuable feedback. We have made substantial revisions and
provided detailed responses to address all concerns, both in this rebuttal and
in the revised version of the manuscript.

Best regards,
Authors of Submission 4398

---

### Author Response · Authors · 2025-11-30
**Summary of Our Responses (1/2)**

Dear Area Chairs and Reviewers,

We sincerely thank you for your constructive and insightful feedback. Below we
provide a consolidated summary of **common concerns** raised across multiple
reviewers, followed by **individual concerns** specific to each reviewer. For
each point, we briefly restate our conclusion and indicate where the
corresponding changes were made in the revised manuscript (highlighted in red).
Detailed, point-by-point responses are provided in the rebuttal to each
reviewer.

---

### Common Concerns

**CC1 – Runtime and memory efficiency of EAMET (ifDP–W1, XmZ3–W2)**

We conducted detailed empirical measurements showing that EAMET incurs nearly
identical runtime and memory overhead compared to MEMIT at various scales. The
additional alignment step introduces only negligible cost. The full results and
discussion are incorporated in **Appendix F.6** of the revised manuscript.


**CC2 – Impact of editing order on EAMET’s performance (ifDP–W2, jhJa–W2)**

We systematically investigate the effect of different editing sequences by
randomly shuffling the order of all knowledge items and evaluating EAMET on
large-scale batches. The results show that EAMET is insensitive to editing
order, exhibiting strong robustness to sequence variations on both CounterFact
and ZsRE. These results are incorporated in **Section 6.3** of the revised
manuscript.


**CC3 – Clarification of presentation and notation (XmZ3–W3, XmZ3–W4, jhJa–W5)**

We have carefully revised the manuscript to clarify all previously ambiguous
points in the presentation, including missing definitions, notation, and figure
references. All corresponding changes are highlighted in red in the revised
version.

---

### Individual Concerns

**IC1 – Editing success across different knowledge categories (XmZ3–W1)**

We added a detailed analysis of EAMET’s performance across different semantic
relation types. Using relation-specific subsets from CounterFact, we compare
EAMET with existing methods and show that EAMET consistently achieves the
highest efficacy and generalization across most relation categories.

We further highlight the **twin-city** relation as a representative case, where
EAMET attains **2× higher efficacy** and **4× higher generalization** than PMET,
while MEMIT nearly fails on this relation. We analyze this phenomenon and show
that it supports our claim that alignment between key and residual embeddings is
crucial when keys are highly overlapping. These results and discussions are
presented in **Figure 8 of Appendix F.1** in the revised manuscript.


**IC2 – Performance under single-edit and small-batch scenarios (QuKf–W1)**

We added experiments on LLaMA2-7B covering single-edit and small-batch
scenarios. The results show that EAMET and MEMIT behave similarly at very small
scales (1–10 edits), but as the number of edited facts increases, EAMET remains
stable while MEMIT rapidly degrades. This demonstrates that EAMET retains strong
performance from small to large batch sizes. The full results are provided in
**Appendix F.7** of the revised manuscript.


**IC3 – Use of KL divergence over similarity-based softmax distributions
(QuKf–W2)**

We clarified the rationale behind using KL divergence over similarity-based
softmax distributions to measure embedding misalignment. In the revised
manuscript, we connect our formulation to prior work in contrastive learning and
distributional alignment, where similarity scores are routinely converted into
probability distributions and optimized via divergence-based objectives. We have
also added appropriate citations and explanations in the paragraph addressing
**QuKf–W2 (A2)** around **Line 308** of the revised version.


**IC4 – Analysis of the hyperparameter ($M$) and hyperparameter sensitivity
(QuKf–W3, QuKf–W4)**

We conducted a detailed empirical study on (i) the choice of the neighborhood
size ($M$) in the MSE alignment term and (ii) the sensitivity of EAMET to its
hyperparameters, including ($\lambda_{\text{KL}}$), ($\lambda_{\text{MSE}}$), and
($M$).

The results show that EAMET is robust to the choice of these hyperparameters:
performance varies only slightly across a wide range of settings and remains
consistently better than baseline methods. We include these analyses in
**Appendix F.9** of the revised manuscript.

---

### Meta-Review · Area_Chair_2rvj · 2026-01-07

**Summary:**

This paper investigates a critical scalability failure of existing massive model editing methods, namely the degradation of editing efficacy when a large number of facts are edited simultaneously. The authors attribute this issue to embedding misalignment between key embeddings (used to locate knowledge) and residual/memory embeddings (used to apply updates). They provide a theoretical analysis connecting this misalignment to reconstruction errors in closed-form editing, and propose EAMET, an alignment-aware extension of MEMIT-style methods that explicitly enforces structural consistency between key and residual embedding spaces via KL-divergence–based distributional alignment and pairwise MSE regularization.

Across reviewers, there is broad agreement that the problem addressed is important and practically relevant, particularly for realistic large-scale editing scenarios (e.g., 10k facts, long prefixes, same-subject interference). The proposed method is viewed as technically sound and well-motivated, and the empirical results consistently demonstrate strong improvements in editing efficacy, robustness, and portability across datasets (CounterFact, ZsRE, Wiki-recent) and multiple LLM families.

The main points of discussion concern (1) computational scalability and deployment cost, (2) robustness to editing order, batch size, and conflicting edits, (3) clarity and justification of the theoretical assumptions and objectives, and (4) method complexity and hyperparameter sensitivity. The rebuttal provides extensive new empirical evidence, complexity analyses, and clarifications that resolve most of these concerns. Remaining disagreements are primarily about the strength of theoretical assumptions and expectations of universality rather than empirical validity.

**Reviewer Concerns:**

Concerns largely addressed by the rebuttal:

Runtime, memory, and scalability overhead (Reviewer ifDP, XmZ3):
The authors provide detailed runtime and GPU memory comparisons showing that EAMET incurs only marginal overhead relative to MEMIT (≈1–6% additional cost), even at 10k edits. A complexity breakdown further shows that key embedding preparation and similarity computation are negligible relative to overall editing cost. These results convincingly address scalability and deployment concerns.

Sensitivity to editing order and sequential optimization (Reviewers ifDP, jhJa):
Extensive experiments with randomized editing orders demonstrate that EAMET is largely order-invariant, with only minor performance fluctuations (≈2% in ZsRE). This directly resolves concerns about sequential bias and confirms robustness to arbitrary edit ordering.

Effectiveness in small-batch and single-edit settings (Reviewer QuKf):
Newly added experiments across batch sizes from 1 to 5,000 edits show that EAMET matches MEMIT in trivial regimes (1–10 edits) and significantly outperforms it once batch size exceeds 100. This clarifies that the alignment mechanism does not harm simpler settings while providing clear benefits at scale.

Hyperparameter sensitivity and choice of neighbors M (Reviewer QuKf):
Ablations over M and alignment weights show that EAMET is relatively insensitive to hyperparameter choices, with performance variations remaining modest and consistently outperforming baselines. The intuition behind performance peaks at intermediate M is well explained.

Robustness under same-subject and conflicting edits (Reviewer jhJa):
The authors provide carefully designed experiments with both identical and paraphrased conflicting facts. Results show that EAMET yields consistent outputs (dominated by the first conflicting fact) without propagating inconsistency to other edits, while maintaining overall performance—addressing concerns about semantic conflicts and robustness.

Empirical justification of embedding misalignment (Reviewers XmZ3, QuKf):
Additional category-level analyses and model-wise key similarity statistics strengthen the empirical link between key–residual entanglement and massive editing failure, supporting the paper’s central hypothesis.

Clarity and presentation issues (Reviewer XmZ3, jhJa):
Missing definitions, unclear abbreviations, and insufficient explanation of key formulas were acknowledged and corrected, with expanded intuitive explanations added to the revised manuscript.

Concerns partially addressed or still remains:

Strength of theoretical assumptions (Reviewer jhJa):
The assumption that individual update vectors can be reconstructed from weighted averages of others in large batches remains idealized. While strong empirical evidence is provided showing reconstruction error vanishes at scale—even for semantically unrelated facts—this does not fully eliminate concerns about theoretical generality. This limitation is acknowledged but not fully resolved.

Conceptual complexity and additional hyperparameters (Reviewer QuKf):
Although robustness is empirically demonstrated, EAMET is undeniably more complex than prior methods. Some reviewers remain cautious about whether this complexity could hinder adoption, despite evidence that tuning is not fragile.

**Reviewer Scores:**

Reviewer ifDP :
After detailed runtime/memory analyses and robustness-to-order experiments, the reviewer’s main concerns are directly addressed.


Reviewer XmZ3:
Requests for category-level analysis, complexity breakdown, and clearer explanation of core formulas were thoroughly addressed.


Reviewer QuKf:
Concerns about small-batch performance, hyperparameters, alignment formulation, and dataset/model inconsistency were extensively answered with new experiments and analyses. The reviewer may increase the score.


Reviewer jhJa (Initial: 6, marginal accept):
Most technical and empirical concerns, including order invariance, conflicting edits, and preservation behavior, were addressed with additional targeted experiments. The reviewer  may keep the score.

---

### Decision · Program_Chairs · 2026-01-26

Accept (Poster)